# Neural Causal Discovery with Learnable Input Noise

## Abstract

Learning causal relations from observational time series with nonlinear interactions and complex causal structures is a key component of human intelligence, and has a wide range of applications. Although neural nets have demonstrated their effectiveness in a variety of fields, their application in learning causal relations has been scarce. This is due to both a lack of theoretical results connecting risk minimization and causality (enabling function approximators like neural nets to apply), and a lack of scalability in prior causal measures to allow for expressive function approximators like neural nets to apply. In this work, we propose a novel causal measure and algorithm using risk minimization to infer causal relations from time series. We demonstrate the effectiveness and scalability of our algorithms to learn nonlinear causal models in synthetic datasets as comparing to other methods, and its effectiveness in inferring causal relations in a video game environment and real-world heart-rate vs. breath-rate and rat brain EEG datasets.

## 1 Introduction

From an early age, humans have a remarkable ability to infer causal relations from pure observations (White & Milne (1997); Scholl & Tremoulet (2000); Buchsbaum et al. (2012)). By observing that a left dot moving towards a right dot and the right dot moves correspondingly (the launching effect, see Michotte (1963)), a human can quickly infer that the left dot *causes* the right dot to move (Scholl & Tremoulet (2000)). In fact, much of our way of thinking is via cause and effect. Causal analysis also permits counterfactual reasoning, answering what would have happened if the cause had happened differently. The learning of causality also constitutes much of the scientific endeavor, for example finding the cause of a certain cancer (Bosch et al. (2002)), or discovering gene regulatory networks (Lozano et al. (2009)). In addition, causality plays a key role in neuroscience (Neves et al. (2008); Seth et al. (2015)), economics (e.g. Granger (1969); Stock & Watson (1989)) and finance (Hiemstra & Jones (1994); Granger et al. (2000)).

The study of causality has a long history, yet the application of neural nets to learning causal relations has been scarce. There have been various works that propose methods to infer causal structures with limited model space (e.g. Granger (1969)) which may not be able to model complex nonlinear causal relations, or propose measures to quantify causal strength (e.g. Schreiber (2000); Janzing et al. (2013)) that may not scale to high-dimensional data. For these, the use of universal function approximators like neural nets may be beneficial in inferring causality, which motivate us to propose causal measures that are not only theoretically founded, but also amenable to the learning of function approximators. On the other hand, in the deep learning community, the learning of causal models has not been prevalent, in part due to the lack of theoretical understanding between learning a prediction model and learning a causal model. This also motivates us to propose causal measures, obtained via learning a prediction model, that can deduce causality.

The contributions of this work are as follows:

- We propose a novel measure to quantify causality from observational data, and an effective algorithm, Causal Inference with Learnable Noise (CILN), to estimate it. It is based on minimizing a learnable noise risk of a prediction model, allowing function approximators such as neural nets to learn complex causal relationships.

- We demonstrate on nonlinear synthetic datasets that our method outperforms other causal measures by a large margin, and can scale to a larger number of time series. We also demonstrate that our models are effective on real-world datasets.

## 2 RELATED WORK

The study of causality has a long history, and has been approached from different perspectives. Pearl (e.g. Pearl (2002; 2009); Pearl et al. (2009)) defines causality in terms of intervention and structural dependence, under the structural equation models (SEM). Granger (Granger (1969); Granger & Newbold (1986)) defines causality via prediction: if the prediction of Y via a linear model can be improved by including the information of X, then X causes Y in the Granger sense. Since its proposal, Granger causality has been widely applied in economics (e.g. Joerding (1986)) and neuroscience (e.g. Deshpande et al. (2009); Seth et al. (2015)). To learn nonlinear causal relations, later works also extend Granger causality to kernel methods (Ancona et al. (2004); Marinazzo et al. (2008a;b); Sindhwani et al. (2012)). To clear up the relations between these two notions of causality, White et al. (2011) provide conditions under which Granger causality can deduce direct structural causality in a general settable system framework (White & Chalak (2009)), with direct structural causality as a natural extension of Pearl causality in settable systems. Our method also utilizes the high-level idea of inferring causal relations via prediction, and building on the work of White et al. (2011) we propose a novel measure that can likely uncover direct structural causality under certain conditions.

Numerous methods have been proposed to discover causal structures from data. One important class is constraint-based methods, for example PC (Spirtes et al. (2000)), rankPC (Harris & Drton (2013)), IC (Pearl (2002)), and FCI (Spirtes et al. (2000)), which require repetitive conditional independence tests. Hybrid methods, e.g. MMHC (Tsamardinos et al. (2006)), require repetitive estimation of conditional association score (e.g. using conditional mutual information). In comparison, under the scope of time series, our method can simultaneously discover multiple variables that directly cause the variable of interest. Score-based methods search for the structure that yields the optimal score w.r.t. the data, generally using greedy search methods, for example GES (Chickering (2002)), rankGES (Nandy et al. (2018)) and GIES (Hauser & Bühlmann (2012)). This in general requires $\Theta(N^2)$ steps ($N$ denoting the number of nodes in the graph), and the number of neighboring states may grow very large at each step. In comparison, our method only requires training $N$ models for causal discovery with time series.

In addition, various measures have been proposed to quantify causality. Schreiber (2000) proposes transfer entropy as a measure for causality. It measures the mutual information between the current Y and the past of X, conditioned on the past of Y, to quantify the directional information transfer. As noted in Marinazzo et al. (2008a), Granger causality as defined in Granger & Newbold (1986) implies nonzero transfer entropy. Janzing et al. (2013) analyze several causal measures, and conclude that they are unsatisfactory measures of causal strength. They then propose causal influence, defined via the KL-divergence between the original joint distribution and the distribution with a set of causal arrows broken. We note that both the calculation of transfer entropy and causal influence requires density estimation of the full joint distributions of the input and output, which could easily become difficult in high dimensions. This motivates us to propose new causal measures based on risk minimization, which is much easier with high input dimensions. Besides, various other works have also proposed methods to infer causal structure under some specific conditions, for example, causal additive noise models (Hoyer et al. (2009)), information-geometric causal inference (Daniusis et al. (2012); Janzing et al. (2012)), dynamic causal modeling (Friston et al. (2003)), etc.

There has also been works that approach causality from a machine learning perspective, or have architectures that put causality to mind. Lopez-Paz et al. (2015) study causal inference as a supervised learning problem, where the pairwise causal directions are given as labels for training. Louizos et al. (2017) utilize a variational autoencoder (VAE) structure to estimate the unknown latent space summarizing the confounders and the causal effect. Kipf et al. (2018) propose a neural relational inference architecture, which simultaneously infers the interactions and learning the dynamics from observational data. Designing causal inference into the model architecture can also be beneficial in certain applications. For example, Kansky et al. (2017) propose a Schema Network that allows re-

gression planning from a goal through causal chains, and demonstrates zero-shot learning in a suite of variations of Atari Breakout games.

Our method also relates to sparse learning/feature selection methods, for example L1 regularization and group L1 regularization (e.g. Meier et al. (2008); Scardapane et al. (2017)). Although L1 or group L1 regularization encourage sparsity in the weights of neural nets, the regularization is model and input dependent, in contrast to our method's invariance to model structure change and inputs rescaling. For example, a three-layer linear network and a collapsed one-layer linear network representing the same function can induce different L1 regularizations. For another example, if one of the input time series is scaled by a factor of 0.1 while other variables remain unchanged, the L1 regularization will be different due to a differently learned model. In comparison, our learnable noise risk when minimized is invariant to the above two kinds of changes, making it especially suitable to discover causality where the scale of different time series may span orders of magnitude, and the model structure may vary, as demonstrated in Section 4.1.

## 3 METHOD

### 3.1 PROBLEM DEFINITION

Although causal data can be inferred from observations at individual time points, we are primarily concerned with time series. Time series have extra structure that is particularly useful for causal inference: causes must precede their effects. Therefore, we consider $N$ time series $x^{(1)}, x^{(2)}, ...x^{(N)}$, where each time series $x^{(i)} = (x_1^{(i)}, x_2^{(i)}, ...x_t^{(i)}, ...)$ and each $x_t^{(i)} \in R^M$ is an $M$-dimensional vector. Denote $X_{t-1}^{(i)} = (x_{t-K}^{(i)}, x_{t-K+1}^{(i)}, ...x_{t-1}^{(i)})$ with maximum time horizon of $K$, and $\mathbf{X}_{t-1} = \{X_{t-1}^{(i)}\}, i = 1, 2, ...N$. We also denote $\mathbf{X}_{t-1}^{(\hat{j})} = \mathbf{X}_{t-1} \backslash X_{t-1}^{(j)}$, i.e. $\mathbf{X}_{t-1}$ excluding $X_{t-1}^{(j)}$, to notationally differentiate with the variable of interest $X_{t-1}^{(j)}$. We assume that the time series is generated by a canonical settable system (White et al. (2011)). We adopt the settable system (White & Chalak (2009)) paradigm due to the following reasons: (1) as a natural extension to SEMs, it facilitates optimization, equilibrium, and learning; (2) it can formally link Granger causality with Pearl causality; (3) it is general enough to encompass a large number of practical scenarios, including time series. Intuitively, each variable in a settable system can either be determined by direct setting (which formalizes interventions), or by a response function on the settings of other variables. A canonical settable system is a settable system where each variable's setting equals its response, allowing the system to evolve naturally without intervention, formalizing time series. In this paper, we also assume *causal sufficiency* (Peters et al. (2017)), i.e., each time series $x^{(i)}$ can only be structurally caused by the time series from $x^{(1)}, x^{(2)}, ...x^{(N)}$, and generated by stationary response functions $h_i$ that are unknown to the learner:

$$
\begin{cases}
x_t^{(1)} := h_1(\mathbf{X}_{t-1}, u_1) \\
x_t^{(2)} := h_2(\mathbf{X}_{t-1}, u_2) \\
... \\
x_t^{(N)} := h_N(\mathbf{X}_{t-1}, u_N)
\end{cases}
\tag{1}
$$

for $t = K + 1, K + 2, ...$ . Here $u_i \in R^M, i = 1, 2, ...N$ are noise variables that are mutually independent, are independent of any $X_{t-1}^{(i)}, x_t^{(i)}, i \in \{1, 2, ...N\}$, and are effective arguments of the response functions $h_i$. Also in this paper, we only consider "causality in mean", i.e. the causal relations, if exists, influence the mean value of other variables. For any $i, j \in \{1, 2, ...N\}$, we assume that the variables $(\mathbf{X}_{t-1}^{(\hat{j})}, X_{t-1}^{(j)}, x_t^{(i)})$ have probability density function $P(\mathbf{X}_{t-1}^{(\hat{j})}, X_{t-1}^{(j)}, x_t^{(i)})$. We will leave time series with hidden variables (therefore confounding can occur) for future work. We note that even without considering hidden variables, Eq. 1 is very general and already encompasses a wide range of scenarios.

In order to state the goal of the learner who wants to learn causality from the observations $x^{(1)}, x^{(2)}, ...x^{(N)}$, we need to rigorously define causality. Here, we restate the definitions of *direct structural causality* (White et al. (2011)) and *Granger causality* (Granger & Newbold (1986)) using our notations of the system Eq. 1, the former being a natural extension to Pearl causality (Pearl (2009)) in the settable systems.

**Direct structural causality** (White et al. (2011)) *We say $X_{t-1}^{(j)}, j \neq i$ does not directly struc-turally cause $x_t^{(i)}$, if for all possible values of $\mathbf{X}_{t-1}^{(\hat{j})}$ and $u_l$, $l \in 1, 2, ...N$, the function $X_{t-1}^{(j)} \rightarrow h_i(\mathbf{X}_{t-1}, u_i)$ is constant in $X_{t-1}^{(j)}$. Otherwise, we say $X_{t-1}^{(j)}$ directly structurally causes $x_t^{(i)}$.*

Granger (1969) defines causality in terms of conditional expectations. Later, Granger & Newbold (1986) define it using conditional distributions. We use the latter definition, since it is more general, and also facilitates connection with direct structural causality, and the algorithms proposed in this paper. In this work, we only consider causality *between* different time series, i.e. requiring $j \neq i$ as default.

**Granger causality** (Granger & Newbold (1986)) *We say $X_{t-1}^{(j)}, j \neq i$ does not Granger-cause $x_t^{(i)}$, if $P(x_t^{(i)}|X_{t-1}^{(j)}, \mathbf{X}_{t-1}^{(\hat{j})}) = P(x_t^{(i)}|\mathbf{X}_{t-1}^{(\hat{j})})$, i.e. the conditional distribution function of $x_t^{(i)}$ given $X_{t-1}^{(j)}, \mathbf{X}_{t-1}^{(\hat{j})}$ is identical to the conditional distribution function of $x_t^{(i)}$ given $\mathbf{X}_{t-1}^{(\hat{j})}$. Otherwise, we say $X_{t-1}^{(j)}$ Granger-causes $x_t^{(i)}$.*

The goal of the learner is, given only $x^{(1)}, x^{(2)}, ...x^{(N)}$, determine for any $x_t^{(i)}$, whether $X_{t-1}^{(j)}$ directly structurally causes $x_t^{(i)}$ for each $j = 1, 2, ...N$.

## 3.2 GRANGER CAUSALITY IMPLIES DIRECT STRUCTURAL CAUSALITY IN EQ. (1)

For our system Eq. (1), applying the results by White et al. (2011), we have that Granger causality is a sufficient condition for direct structural causality. See Appendix A for the detailed theorem and proof.

*For system Eq. 1, for any $i, j \in \{1, 2, ...N\}, i \neq j$, if $X_{t-1}^{(j)}$ Granger-causes $x_t^{(i)}$, then $X_{t-1}^{(j)}$ directly structurally causes $x_t^{(i)}$.*

The reason that here Granger causality can deduce direct structural causality is in part due to the fact that for system Eq. (1), conditional exogeneity (White et al. (2011)) is automatically satisfied by the assumptions of the system.

Note that the reverse of the statement is not true, i.e. Granger non-causality does not necessarily imply direct structural non-causality (White & Lu (2010) give several examples). They also note that these instances are exceptional, in that mild perturbations to their structures destroy the Granger non-causality.

## 3.3 OUR METHOD

Based on section 3.2, if we have an algorithm that can deduce Granger causality in system Eq. (1), then we can immediately deduce direct structural causality. In general, a direct test of Granger causality is difficult. In particular, when the number of time series is large or the dimension $M$ of each time series is large, density or mutual information-based methods like transfer entropy ($I(x_t^{(i)}; X_{t-1}^{(j)}|\mathbf{X}_{t-1}^{(\hat{j})})$, see Schreiber (2000)), may not give a good estimate. Alternatively, various works have resorted to testing whether the prediction of $x_t^{(i)}$ given $X_{t-1}^{(j)}, \mathbf{X}_{t-1}^{(\hat{j})}$ is better than given only $\mathbf{X}_{t-1}^{(\hat{j})}$, under some limited functional space. For example, in his original work, Granger (1969) investigates causality with linear function predictors. Later works have extended it to kernel methods, e.g., Ancona et al. (2004); Marinazzo et al. (2008a;b); Sindhwani et al. (2012), which essentially estimate linear Granger causality on the feature space of the kernel.

To enable causal learning with potentially highly nonlinear response functions, it may be desirable to use universal function approximators (Hornik (1991)) such as neural nets. As the main contribution of this paper, we will provide a novel causal measure and corresponding algorithm to estimate it. The measure is based on optimizing an objective containing a function approximator, thus allowing neural nets to apply.

Our algorithm is inspired by asking a counterfactual question during the learning of a prediction model. Specifically, it asks:

---

**Algorithm 1 Causal Inference with Learnable Noise (CILN)**

---

**Require** $x_t^{(i)}, \mathbf{X}_{t-1}$, for $i \in \{1, 2, ...N\}, t \in \mathbf{T} = \{K+1, K+2, ...\}$.
**Require** $\eta_0$: a small value for initialization of $\boldsymbol{\eta}$.
**Require** $\lambda$: coefficient for the mutual information term.
1: **for** $i$ in $\{1, 2, ...N\}$ **do:**
2:      Initialize function approximator $f_\theta$.
3:      Initialize $\boldsymbol{\eta} = (\eta_1, \eta_2, ...\eta_N) = (\eta_0\mathbf{1}, \eta_0\mathbf{1}, ...\eta_0\mathbf{1})$, where each element $\eta_0\mathbf{1}$ is a $KM-$
         dimensional vector, same dimension as $X_{t-1}^{(j)}$.
4:      $(f_{\theta^*}, \boldsymbol{\eta}^*) \leftarrow \text{Minimize}_{(f_\theta, \boldsymbol{\eta})} \hat{R}_{\mathbf{X}, x^{(i)}, \boldsymbol{\epsilon}}[f_\theta, \boldsymbol{\eta}]$ (Eq. 4) with e.g. gradient descent
5:      $W_{ji} \leftarrow I(\tilde{X}_{t-1}^{(j)(\eta_j^*)}; X_{t-1}^{(j)})$, for $j = 1, 2, ...N$.
6: **end for**
7: **return** $W$

---

*How much noise can I add to $X_{t-1}^{(j)}$, without making the best prediction of $x_t^{(i)}$ worse?*

To give a quantitative answer to this question, we define a *learnable noise risk*:

$$R_{\mathbf{X}, x^{(i)}}[f_\theta, \boldsymbol{\eta}] = \mathbb{E}_{\mathbf{X}_{t-1}, x_t^{(i)}, \boldsymbol{\epsilon}} \left[ \left( x_t^{(i)} - f_\theta(\tilde{\mathbf{X}}_{t-1}^{(\boldsymbol{\eta})}) \right)^2 \right] + \lambda \cdot \sum_{j=1}^N I(\tilde{X}_{t-1}^{(j)(\eta_j)}; X_{t-1}^{(j)}) \quad (2)$$

where $\tilde{\mathbf{X}}_{t-1}^{(\boldsymbol{\eta})} := \mathbf{X}_{t-1} + \boldsymbol{\eta} \odot \boldsymbol{\epsilon}$ (or element-wise, $\tilde{X}_{t-1}^{(j)(\eta_j)} := X_{t-1}^{(j)} + \eta_j \cdot \epsilon_j$, $j = 1, 2, ...N$) is the noise-corrupted inputs with *learnable* noise amplitudes $\eta_j \in R^{KM}$, and $\epsilon_j \sim N(\mathbf{0}, \mathbf{I})$. $\lambda > 0$ is a positive hyperparameter for the mutual information $I(\cdot, \cdot)$. Intuitively, the minimization of the second term $I(\tilde{X}_{t-1}^{(j)(\eta_j)}; X_{t-1}^{(j)})$ requires the noise amplitude $\eta_j$ to go up. The minimization of the first term requires the noise amplitude $\eta_j$ to go down, and the larger causal strength from $X_{t-1}^{(j)}$ to $x_t^{(i)}$, the larger this force. The minimization of the two terms strikes a balance, at which point the $I(\tilde{X}_{t-1}^{(j)(\eta_j)}; X_{t-1}^{(j)})$ measures how many bits of information does time series $j$ need to provide to the learner, without making the prediction worsened. Thus, we propose to use

$$W_{ji} = I(\tilde{X}_{t-1}^{(j)(\eta_j^*)}; X_{t-1}^{(j)}) \quad (3)$$

as another measure for causality, where $(f_{\theta^*}, \boldsymbol{\eta}^*) = \text{argmin}_{(f_\theta, \boldsymbol{\eta})} R_{\mathbf{X}, x^{(i)}}[f_\theta, \boldsymbol{\eta}]$ [1]. In Appendix B, we analyze the qualitative and quantitative properties of the learnable noise risk, and give intuitions why it is likely to select the variables that directly structurally causes $x_t^{(i)}$.

Empirically, we minimize the following empirical risk:

$$\hat{R}_{\mathbf{X}, x^{(i)}, \boldsymbol{\epsilon}}[f_\theta, \boldsymbol{\eta}] = \frac{1}{|\mathbf{T}|} \sum_{t \in \mathbf{T}} \left( x_t^{(i)} - f_\theta(\tilde{\mathbf{X}}_{t-1}^{(\boldsymbol{\eta})}) \right)^2 + \lambda \sum_{j=1}^N I(\tilde{X}_{t-1}^{(j)(\eta_j)}; X_{t-1}^{(j)}) \quad (4)$$

In general, it may be difficult to estimate the mutual information $I(\tilde{X}_{t-1}^{(j)(\eta_j)}; X_{t-1}^{(j)})$ with large dimension of $X_{t-1}^{(j)}$ such that the expression is also differentiable w.r.t. $\eta_j$. Utilizing the property of Gaussian channels, in Appendix C we prove that $I(\tilde{X}_{t-1}^{(j)(\eta_j)}; X_{t-1}^{(j)}) \leq \frac{1}{2} \sum_{l=1}^{KM} \log \left( 1 + \frac{\text{Var}(X_{t-1,l}^{(j)})}{\eta_{j,l}^2} \right)$, where $l$ denotes the $l^{\text{th}}$ element of a vector, and $\text{Var}(X_{t-1,l}^{(j)})$ is the variance of $X_{t-1,l}^{(j)}$ across $t$. Therefore, in practice to improve efficiency, we can optimize an *upper bound* of the learnable noise risk:

$$\hat{R}_{\mathbf{X}, x^{(i)}, \boldsymbol{\epsilon}}^{\text{upper}}[f_\theta, \boldsymbol{\eta}] = \frac{1}{|\mathbf{T}|} \sum_{t \in \mathbf{T}} \left( x_t^{(i)} - f_\theta(\tilde{\mathbf{X}}_{t-1}^{(\boldsymbol{\eta})}) \right)^2 + \frac{\lambda}{2} \sum_{j=1}^N \sum_{l=1}^{KM} \log \left( 1 + \frac{\text{Var}(X_{t-1,l}^{(j)})}{\eta_{j,l}^2} \right) \quad (5)$$

---

[1] Note also that throughout this paper, when talking about causal matrices, the $(j, i)^{\text{th}}$ element always denotes the causal strength from $j$ to $i$.

When the dimension of $X_{t-1}^{(j)}$ is large, differentiable estimate of the mutual information, such as MINE (Belghazi et al. (2018)), can be applied. We provide Algorithm 1 to empirically estimate $W_{ji}$, which we term Causal Inference with Learnable Noise (CILN).

## 4 EXPERIMENTS

To demonstrate that our proposed method works, we test it on both synthetic and real datasets. We first use synthetic datasets, where we know the underlying causal structure and compare with previous causal measures. We then test whether our algorithm can infer causal structure from watching an agent playing video games. Finally, we apply our algorithm to real-world heart-rate vs. breath-rate and rat EEG datasets to test its effectiveness. We use the $\hat{R}_{\mathbf{X},x^{(i)},\boldsymbol{\epsilon}}^{\text{upper}}[f_\theta, \boldsymbol{\eta}]$ (Eq. 5) for optimization for all experiments. The metrics we use are the standard metrics of area under the precision-recall curve (AUC-PR) (Davis & Goadrich (2006)), and area under the ROC curve (AUC-ROC).

### 4.1 SYNTHETIC EXPERIMENT WITH LOG-NORMAL CAUSAL STRENGTHS

In this experiment, we evaluate our method together with other methods with a nonlinear synthetic dataset generated to have a known causal structure (hidden to the methods being compared). We study how they perform with varying number $N$ of time series, with $N$ up to 30. To generate the data, we let each $x_t^{(i)}$ have dimension $M = 1$, and also set the maximum time horizon $K = 3$, so each $X_{t-1}^{(j)}$ is a $K \times M = 3 \times 1$ matrix. We use the following realization of the response function $h_i$ in Eq. (1):

$$x_t^{(i)} = h_i(\mathbf{X}_{t-1}, u_t) = \mathrm{H}_1\left(\sum_{j=1}^N \left[ A_{ji} \odot \mathrm{H}_2(B_j \odot X_{t-1}^{(j)}) \right]\right) + u_t, \ \ i = 1, 2, ...N \quad (6)$$

where $u_t \sim N(\mathbf{0}, \mathbf{I}) \in R^M$, $\odot$ denoting element-wise multiplication, and $\mathrm{H}_1$ and $\mathrm{H}_2$ are two nonlinear functions to make the response functions nonlinear. In this experiment, we use $\mathrm{H}_1(x) = \mathrm{softplus}(x) = \log(1 + e^x)$, and $\mathrm{H}_2(x) = \tanh(x)$, we also find similar performance with other choices of nonlinear functions. $B_j$ is a $K \times M$ random matrix, whose element is sampled from $U[-1, 1]$. $A_{ji}$ is a $K \times M$ matrix, with 0.5 probability of being a zero matrix and 0.5 probability of being a nonzero random matrix, characterizing the underlying causal strength from $j$ to $i$. Crucially, to reflect that the causal strength may span different orders of magnitude, if $A_{ji}$ is sampled to be a nonzero matrix, then the amplitude of each of its element is sampled from a log-normal distribution with $\mu = 1, \sigma = 0$, their sign sampling from $U\{-1, 1\}$. Denote $A_{indi}$ as the 0-1 indicator matrix of causality ($A_{indi,ji} = 1$ if $|A_{ji}| > 0$; 0 otherwise). The goal of each algorithm being evaluated is to produce an $N \times N$ causal matrix $\tilde{A}$, where each entry $\tilde{A}_{ji}$ characterizes the causal strength from $j$ to $i$. Then the flattened $\tilde{A}$ is evaluated against the flattened $A_{indi}$ (excluding diagonal elements of the matrices), producing the metrics of AUC-PR and AUC-ROC.

In general, for a large $N$, the number of possible causal graphs grows double exponentially: there are $2^{N^2}$ possible matrix of $A_{indi}$. To give an estimate, for $N = 3, 4, 5, 8, 10, 20, 30$, there are $512, 6.6 \times 10^4, 3.3 \times 10^7, 1.8 \times 10^{19}, 1.2 \times 10^{30}, 2.6 \times 10^{120}, 8.5 \times 10^{270}$ number of possible graphs, respectively. Therefore, estimating the causal graph is in general a non-trivial task when $N$ is large. We compare our algorithm, with previous methods including transfer entropy (Schreiber (2000)), causal influence (Janzing et al. (2013)), linear Granger causality (Granger (1969); Ding et al. (2006)), and a baseline of mutual information $\tilde{A}_{ji} = I(X_{t-1}^{(j)}; x_t^{(i)})$ (which gives $\tilde{A}_{ji} = \tilde{A}_{ij}$). The implementation details for each method and each experiment are provided in Appendix D and Appendix E, respectively. Table 1 and 2 shows the average AUC-PR and AUC-ROC with each $N$, each with four random initializations of the true underlying causal matrix $A$ and dataset.

We see that not only does our method outperform other methods by a large margin across all $N$s, it also shows good performance when $N$ is as large as 30, demonstrating our method's capability and scalability to infer complex causal structures from interacting time series. For the Causal Influence method, although it has very good mathematical properties, it may be impractical in practice, as is also shown in the table. This is due to that it is defined as the KL-divergence between $(\mathbf{X}_{t-1}, x_{t-1}^{(i)})$

| method | N | 3 | 4 | 5 | 8 | 10 | 15 | 20 | 30 |
|---|---|---|---|---|---|---|---|---|---|
| Ours | | **0.9757** | **0.9820** | **0.9604** | **0.9424** | **0.9302** | **0.9266** | **0.8614** | **0.7740** |
| Transfer Entropy | | 0.9401 | 0.9308 | 0.9059 | 0.7376 | 0.6764 | 0.6197 | 0.5723 | 0.4818 |
| Mutual Information | | 0.9139 | 0.9318 | 0.8775 | 0.8292 | 0.8058 | 0.7687 | 0.7147 | 0.6980 |
| Linear Granger | | 0.7262 | 0.9431 | 0.8054 | 0.8062 | 0.7566 | 0.6589 | 0.6643 | 0.5071 |
| Causal Influence | | 0.7910 | 0.7068 | 0.5481 | 0.3693 | 0.4149 | 0.4798 | 0.4612 | 0.4254 |

Table 1: Average AUC-PR vs. $N$, with random sampling of $A_{indi}$. Bold font marks the top method for each $N$.

| Method | N | 3 | 4 | 5 | 8 | 10 | 15 | 20 | 30 |
|---|---|---|---|---|---|---|---|---|---|
| Ours | | **0.9722** | **0.9580** | **0.9525** | **0.9406** | **0.9382** | **0.9251** | **0.8546** | **0.7810** |
| Transfer Entropy | | 0.8854 | 0.8469 | 0.9035 | 0.7893 | 0.7731 | 0.6609 | 0.6006 | 0.5275 |
| Mutual Information | | 0.8889 | 0.8817 | 0.8611 | 0.8414 | 0.8403 | 0.7711 | 0.7380 | 0.7156 |
| Linear Granger | | 0.6806 | 0.8969 | 0.7787 | 0.8166 | 0.7881 | 0.6887 | 0.6777 | 0.5840 |
| Causal Influence | | 0.7083 | 0.6334 | 0.6077 | 0.4155 | 0.4705 | 0.5707 | 0.5153 | 0.5218 |

Table 2: Average AUC-ROC vs. $N$, with random sampling of $A_{indi}$. Bold font marks the top method for each $N$.

and $(\mathbf{X}_{t-1}^{(\tilde{j})}, x_{t-1}^{(i)})$, each of which is an $(NK + 1)M-$dimensional vector, which can quickly go to high dimensions, where density estimation required to calculate KL-divergence is in general data-hungry and difficult. In comparison, our method that estimates causal strength via minimizing prediction errors is comparatively easier in high dimensions (only have to predict a $M$ dimensional vector conditioned on the inputs), which contributes to a better performance when $N$ is large.

Since in practice, we do not know the underlying causal structure *a priori*, it presents a greater challenge to select the model capacity for $f_\theta$, as compared with supervised learning method where we can do cross-validation. To see how the capacity of the function approximator $f_\theta$ influences our method, we vary the number of layers and the number of neurons in each layer at $N = 10$. Table 3 summarizes the result. We see that our method's performance here is hardly influenced by the model capacity, with only a slight degradation at very low capacity. This shows that our method is quite tolerant and stable with model capacity variations.

| Neurons in hidden layers | AUC-PR | AUC-ROC |
|---|---|---|
| (8) | 0.8969 | 0.9086 |
| (8, 8) | 0.9309 | 0.9388 |
| (16, 16) | 0.9404 | 0.9455 |
| (8, 8, 8) | 0.9339 | 0.9410 |
| (16, 16, 16) | 0.9284 | 0.9288 |
| (8, 8, 8, 8) | 0.9312 | 0.9350 |
| (16, 16, 16, 16) | 0.9199 | 0.9202 |

Table 3: Average AUC-PR and AUC-ROC for different network structures for $N = 10$ with our method. Here for example, (8, 8, 8) means that the $f_\theta$ has 3 hidden layers, each with 8 neurons.

## 4.2 Experiments with Video games

To see how our method can infer causal relations in real videos games, and potentially helping reinforcement learning (RL) or imitation learning (IL), we apply our method to the causal inference between the trajectories of different objects from a trained CNN RL-agent playing Atari Breakout

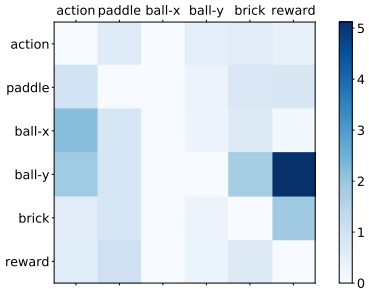

Figure 1: Causal strength $W_{ji}$ inferred by our method with watching a trained CNN playing Breakout. The $(j, i)$ element denotes the inferred causal strength from $j$ to $i$.

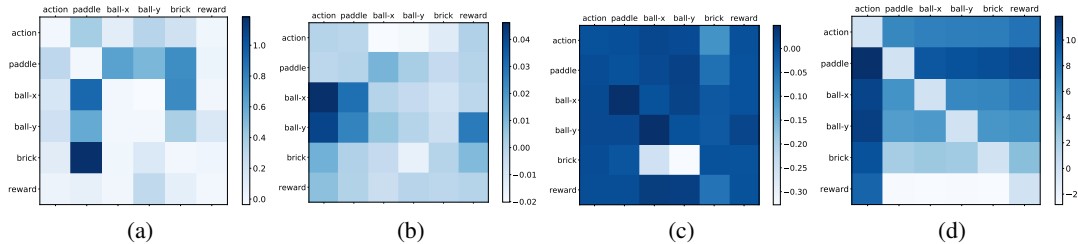

Figure 2: Causal strength inferred by (a) mutual information (b) transfer entropy (c) linear Granger (d) causal influence

.

games[2] (Bellemare et al. (2013), implementation details see Appendix F). Fig 1 shows the inferred $W_{ji}$ matrix, with the $(j, i)^{\text{th}}$ element denoting the inferred causal strength from $j$ to $i$. We see that there is a prominent causal direction from the ball's $y$ position to the reward, which correctly summarizes that the ball's $y$ position has a large influence on the reward. Additionally, the other discovered causal directions with causal strength greater than 1 include: brick $\to$ reward, ball-y $\to$ brick, reward $\to$ paddle, ball-x $\to$ action, ball-y $\to$ action. The former two correctly summarize causal chains from ball-y $\to$ brick $\to$ reward, the latter two show that the ball's $x$ and $y$ positions also have big causal influences on the trained agent's action: in order that the ball does not fall to the bottom, the agent has to position itself at the right position depending on the $x$ and $y$ positions of the ball.

In comparison, mutual information (Fig. 2 (a)) gives a symmetric matrix that does not differentiate the two possible causal directions. Moreover, it discovers a high mutual information between brick and paddle, which does not related to the underlying causal mechanism, and also missed the causal arrows ball-y→brick→reward. For transfer entropy, although it discovers a number of prominent causal arrows (e.g. ball-x→action, ball-y→action, ball-y→reward, brick→reward), it also gives relative high scores for some incorrect causal arrows: brick→ action, ball-y→ball-x, and missed an important causal arrow ball-y→brick. For linear Granger, it gives most arrows 0 or negative score, failing to discover any useful causal arrows. For causal influence, it also fails to discover useful causal arrows.

The inferred causal matrix and learned models may be useful for downstream learning in RL/IL. The learned models $f_\theta$ can serve as a succinct subset of the environment model $p(s', r | s, a)$ that predicts future state $s'$ and reward $r$ based on current state $s$ and action $a$. Moreover, the agent can utilize the causal matrix to learn policy much more quickly, by reasoning with a causal chain from the reward backward, and from the action forward and backward, and focusing learning policies that can maximally influence the entities in the causal chains. Take the current Breakout game for example. After discovering the causal matrix (Fig. 1) by watching a teacher agent playing, the agent can reason from the reward backward on the causal chain ball-y→brick→reward, and understand that in order to maximize its reward, it has to influence the ball's y position and the number of

---

[2]A video showing the game playing can be seen at https://goo.gl/XGzppc

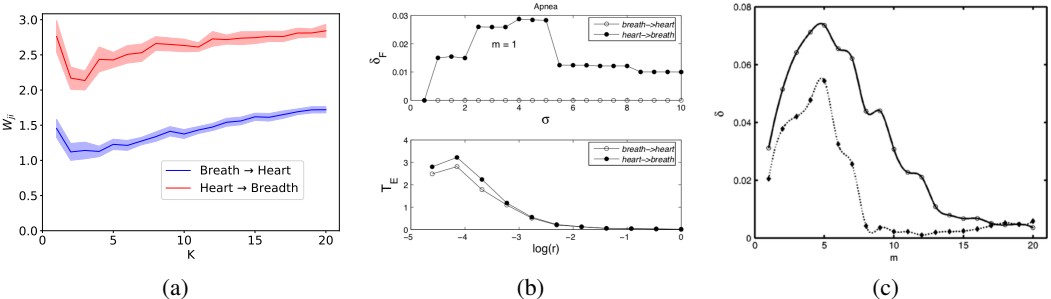

Figure 3: (a) Causal strength $W_{ji}$ inferred by our method with the heart-rate vs. breath-rate dataset, averaged over 50 initializations of $f_\theta$. The shaded areas in this figure and Fig. 4 are the 95% confidence interval. (b) Upper: the filtered causality index vs. varying width of Gaussian kernel $\sigma$ (Marinazzo et al. (2008a)); lower: transfer entropy vs. $r$, the length scale (Schreiber (2000)); (c) The causality index for breath→heart (lower) and heart→breath (upper) in Ancona et al. (2004).

bricks. Reasoning from the action backward on causal chains ball-x→action and ball-y→action, it understands that the teacher agent can obtain high rewards by making actions based on the ball's x and y position (there are no causal chains from the action forward in Fig. 1 so the agent as to learn on its own). The agent can then utilize the above reasoning to develop policies $\pi(a|s)$ where the state $s$ can focus on the ball's x and y position instead of the entire environment state, and the action $a$ can focus on those that can maximally influence not only the reward, but also the ball-y and brick that is on the reward's backward causal chain. This significantly reduces the search space for the policies, and the agent may probably require significantly less number of episodes to learn a good policy. It will be an exciting future research direction to incorporate this causal reasoning into RL and IL to improve sample efficiency.

### 4.3 HEART-RATE VS. BREATH-RATE AND RAT BRAIN EEG DATASETS

Now we test our algorithm with real-world datasets. As a common dataset studied in previous causal works, we use the time-series of the breathing rate and instantaneous heart rate of a sleeping patient suffering from sleep apnea (samples 2350-3550 of data set B from Santa Fe Institute time series contest held in 1991, available in PhysioNet). We apply our method to infer the causal strength between the breathing rate and heart rate, with different maximum time horizon $K$. The result is shown in Fig. 3. The causal strength from heart to breath is significantly higher than the reverse direction, consistent with the results from previous causal inference methods (Schreiber (2000); Ancona et al. (2004); Marinazzo et al. (2008a)) as also shown in Fig. 3. Notably, the causal strength remains at roughly the same level for different $K$s, in contrast to the decaying causality index w.r.t. increasing history length in (Ancona et al. (2004), Fig. 3 (c)) showing a merit of our method in estimating causal strength across different time-horizons, aided by the flexibility of $f_\theta$ in extracting the right information to predict the future. The implementation details in this section is provided in Appendix G.

As a second real-world example, we apply our algorithm in estimating the causal strength of the EEG signals between the right and left cortical intracranial electrodes (Quiroga), also studied in Ancona et al. (2004); Quiroga et al. (2002); Marinazzo et al. (2008a). Figure 4 (left) shows the inferred causal strength $W_{ji}$ for the EEG signals of a normal rat. We see that there is only a slight asymmetry, with the right channel having a slightly stronger influence on the left channel than the reverse direction. Figure 4 (right) shows $W_{ji}$ for the EEG signals with unilateral lesion in the rostral pole of the reticular thalamic nucleus. We see that there is stronger causal influence from the left to the right channels. Compared with the result of previous works Ancona et al. (2004); Marinazzo et al. (2008a) as also shown in Fig. 7, we see that all methods correctly infer the causal relations before and after brain lesion. In addition, our method shows non-decaying causal strength with increasing history length, in contrast to the decaying causality index in Ancona et al. (2004), again demonstrating our method's insensitivity against history length, due to its flexibility in extracting the right amount of information in order to predict the future. The above two applications demonstrate our method's capability in inferring the causal relations from noisy, real-world data.

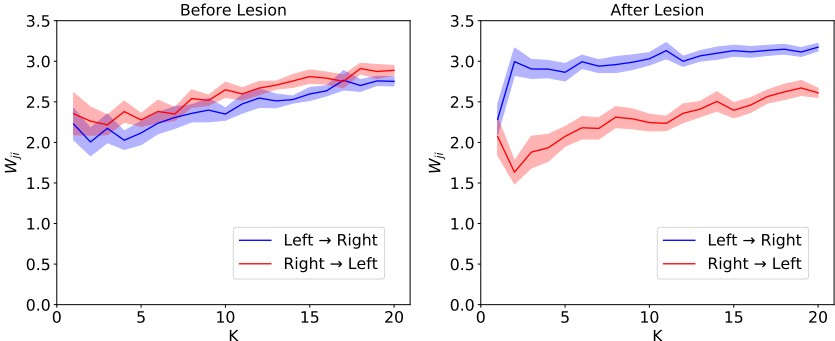

Figure 4: Causal strength inferred by our method with the EEG datasets, for different maximum time horizon $K$, averaged over 50 initializations of $f_\theta$. (Left) Causal strength $W_{ji}$ for the EEG signal for a normal rat. (Right) Causal strength $W_{ji}$ of EEG signal from the same rat, after brain lesion.

## 5 DISCUSSION AND CONCLUSION

In this paper, we have addressed causal inference by proposing a novel causal measure, defined via a novel *learnable noise risk*, that allows function approximators like neural nets to learn complex causal relations via the Causal Inference with Learnable Noise (CILN) algorithm. We provide intuitions that the algorithm is likely to discover variables that directly structurally cause the variable of interest. We demonstrated in synthetic nonlinear datasets that our method outperforms previous methods by a large margin in inferring causal relations with complex structures, can scale to a large number of time series, and is hardly influenced by the capacity of the function approximator. Our method also correctly infers the causal arrows by watching a trained CNN playing Breakout, and give causal directions consistent with prior works in real-world heart-rate vs. breath-rate and rat EEG datasets.

Our CILN algorithm provides many opportunities for downstream tasks. As discussed in Section 4.2, we can incorporate CILN into reinforcement learning or imitation learning, and by reasoning backward and forward with the discovered causal chains, the agent may be able to learn useful policies with much less number of episodes. It may also help interpret the learned neural nets, by quantifying the causal strength from the input to different hidden neurons to the output neurons. Since $W_{ji}$ is based on mutual information which is scale-free, it may provide additional useful information on the internal mechanisms of neural net, in addition to the weights and gradients. CILN can also help decipher complex real-world systems, for example in neuroscience, economics and finance. As for the error effect of the algorithm, it depends on how the inferred causal matrix is utilized qualitatively or quantitatively downstream. For application in RL/IL, a falsely discovered causal arrow will increase the policy search space and thus reduce sample efficiency, but may not influence final performance since the RL/IL algorithm will eventually learn a (sub)optimal policy within the larger search space. Missed causal arrows (false negatives) may lead to the agent's negligence of certain good policies, therefore it is important for the RL/IL algorithm to always allow a certain amount of exploration in addition to focusing policies on the causal chain. For helping interpretability of neural nets and deciphering complex real-world systems, the false positives/negatives of causal relations will influence the qualitative understanding and potential decision making downstream, and it is important to set a higher/lower threshold for the causal score depending on whether false positives or negatives are deemed less desirable.

Above all, we believe our work can not only pave the way for future exciting advancements in enabling machine learning models to understand causality, a key component of human intelligence, but also endow researchers with a useful tool for deciphering the causal relations in complex systems.

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

# Appendix

## A  THEOREM 1 AND PROOF

Here, we formally propose a theorem corresponding to the statement in section 3.2, and provide proof afterwords.

**Theorem 1.** *For system Eq. 1, for any $i, j \in \{1, 2, ...N\}, i \neq j$, if $X_{t-1}^{(j)}$ Granger-causes $x_t^{(i)}$, then $X_{t-1}^{(j)}$ directly structurally causes $x_t^{(i)}$.*

*Proof.* We base the proof on the Theorem 5.6 in White et al. (2011). Firstly, by definition, the system Eq. (1) belongs to the canonical settable system (Def. 3.3 in White et al. (2011)), on which their Theorem 5.6 is based. To prove that in our system Granger causality can deduce direct structural causality, we only have to prove that the assumption A.1 and assumption A.2 in White et al. (2011) are satisfied by our system. If we identify our $x_t^{(i)}$ with their $Y_{1,t}$, our $\mathbf{X}_{t-1}$ with their $\mathbf{Y}_{t-1}$, our $x_t^{(j)}$ with their $Y_{2,t}$, our $u_{i,t}$ (our $u_i$ at time $t$) with their $U_{1,t}$, our $u_{j,t}$ with their $U_{2,t}$, their $\mathbf{Z}_t = \varnothing$, $\mathbf{W}_t = \varnothing$, then our system Eq. (1) satisfies their Assumption A.1. Additionally, by definition, our $u_i \in R^M, i = 1, 2, ...N$ are random variables that are mutually independent, and also independent of any $X_{t-1}^{(i)}, x_t^{(i)}, i \in \{1, 2, ...N\}$. Therefore, our system satisfies their strict exogeneity $(\mathbf{Y}_{t-1}, \mathbf{Z}_t) \perp\!\!\!\perp U_{1,t}$ (in our representation $(\mathbf{X}_{t-1}, \varnothing) \perp\!\!\!\perp u_{i,t}$), which is a sufficient condition for Assumption A.2. Therefore, both their Assumption A.1 and Assumption A.2 are satisfied by our system Eq. (1). Applying their Theorem 5.6, we prove Theorem 1.

∎

## B  ANALYZING THE PROPERTY OF THE LEARNABLE NOISE RISK

Firstly we state the assumption that will be used throughout this section:

**Assumption 1.** *Assume that $f_\theta \in \mathcal{F}$ is a continuous function and has enough capacity so that it can approximate any $\int dx_t^{(i)} P(x_t^{(i)}|\mathbf{X}_{t-1}) x_t^{(i)}$. Let $j \neq i$ and assume that $P(X_{t-1}^{(j)})$ has support with intrinsic dimension of $KM$.*

Also we emphasize that in this paper, the expected risks (with symbol $R$) are w.r.t. the distributions, and the empirical risks (with symbol $\hat{R}$) are w.r.t. a dataset drawn from the distribution, with finite number of examples. The theorems in this paper are all proved w.r.t. distributions (assuming infinite number of examples). Sample complexity results will be left for future work.

The structure of this section is as follows. First in subsection B.1 and B.2, we prove three lemmas that will be helpful for the following analysis. Then in subsection B.3, we analyze the property of the learnable noise risk both qualitatively and quantitatively, and argue why the learnable noise risk is likely to select the variables that directly structurally causes $x_t^{(i)}$.

### B.1  PROVING A LEMMA

Here we prove Lemmas 1.1.

**Lemma 1.1.** *Suppose that Assumption 1 holds, we have*

$$argmin_{f_\theta} R_{\mathbf{X},x^{(i)}}[f_\theta] = \int dx_t^{(i)} P(x_t^{(i)}|\mathbf{X}_{t-1}) x_t^{(i)} \tag{7}$$

*and*

$$min_{f_\theta} R_{\mathbf{X},x^{(i)}}[f_\theta] = \mathbb{E}_{\mathbf{X}_{t-1},x_t^{(i)}} \left[ \left( x_t^{(i)} - \int dx_t^{(i)} P(x_t^{(i)}|\mathbf{X}_{t-1}) x_t^{(i)} \right)^2 \right] \tag{8}$$

*In other words, for the MSE risk, its minimum is attained when $f_\theta(\mathbf{X}_{t-1})$ is the expectation of $x_t^{(i)}$ conditioned on $\mathbf{X}_{t-1}$.*

*Proof.* The proof of the lemma is adapted from Papoulis (1985). The risk

$$
\begin{aligned}
R_{\mathbf{X},x^{(i)}}[f_\theta] &= \mathbb{E}_{\mathbf{X}_{t-1},x_t^{(i)}} \left[ \left( x_t^{(i)} - f_\theta(\mathbf{X}_{t-1}) \right)^2 \right] \\
&= \int d\mathbf{X}_{t-1} dx_t^{(i)} \cdot P(\mathbf{X}_{t-1}, x_t^{(i)}) \left( x_t^{(i)} - f_\theta(\mathbf{X}_{t-1}) \right)^2 \\
&= \int d\mathbf{X}_{t-1} P(\mathbf{X}_{t-1}) \int dx_t^{(i)} P(x_t^{(i)}|\mathbf{X}_{t-1}) \left( x_t^{(i)} - f_\theta(\mathbf{X}_{t-1}) \right)^2
\end{aligned}
$$

Note that here $(x_t^{(i)} - f_\theta(\mathbf{X}_{t-1}))^2 \equiv \left\langle x_t^{(i)} - f_\theta(\mathbf{X}_{t-1}), x_t^{(i)} - f_\theta(\mathbf{X}_{t-1}) \right\rangle$ is an inner product in $R^M$.

For any $\mathbf{X}_{t-1}$, treating $f_\theta(\mathbf{X}_{t-1}) \in R^M$ as a vector, let's calculate its value such that the integral $F(f_\theta(\mathbf{X}_{t-1})) := \int dx_t^{(i)} P(x_t^{(i)}|\mathbf{X}_{t-1}) \left( x_t^{(i)} - f_\theta(\mathbf{X}_{t-1}) \right)^2$ attains its minimum.

Let

$$
\begin{aligned}
0 &= \frac{\partial}{\partial f_\theta(\mathbf{X}_{t-1})} F(f_\theta(\mathbf{X}_{t-1})) \\
&= \frac{\partial}{\partial f_\theta(\mathbf{X}_{t-1})} \int dx_t^{(i)} P(x_t^{(i)}|\mathbf{X}_{t-1}) \left( x_t^{(i)} - f_\theta(\mathbf{X}_{t-1}) \right)^2 \\
&= -2 \int dx_t^{(i)} P(x_t^{(i)}|\mathbf{X}_{t-1}) \left( x_t^{(i)} - f_\theta(\mathbf{X}_{t-1}) \right)
\end{aligned}
$$

we have

$$
\begin{aligned}
\int dx_t^{(i)} P(x_t^{(i)}|\mathbf{X}_{t-1}) x_t^{(i)} &= \int dx_t^{(i)} P(x_t^{(i)}|\mathbf{X}_{t-1}) f_\theta(\mathbf{X}_{t-1}) \\
&= f_\theta(\mathbf{X}_{t-1}) \int dx_t^{(i)} P(x_t^{(i)}|\mathbf{X}_{t-1}) \\
&= f_\theta(\mathbf{X}_{t-1})
\end{aligned}
$$

Therefore, for any $\mathbf{X}_{t-1}$, $f_\theta(\mathbf{X}_{t-1}) = \int dx_t^{(i)} P(x_t^{(i)}|\mathbf{X}_{t-1}) x_t^{(i)}$ is the only stationary point for $F(f_\theta(\mathbf{X}_{t-1}))$.

Taking the second derivative, we have

$$
\frac{\partial^2}{(\partial f_\theta(\mathbf{X}_{t-1}))^2} F(f_\theta(\mathbf{X}_{t-1})) = 2 \int dx_t^{(i)} P(x_t^{(i)}|\mathbf{X}_{t-1}) \mathbf{I} = 2\mathbf{I}
$$

where $\mathbf{I}$ is an $M \times M$ identity matrix, which is always positive definite.

Therefore, for any $\mathbf{X}_{t-1}$, $f_\theta(\mathbf{X}_{t-1}) = \int dx_t^{(i)} P(x_t^{(i)}|\mathbf{X}_{t-1}) x_t^{(i)}$ is the only global minimum of $F(f_\theta(\mathbf{X}_{t-1}))$ w.r.t. $f_\theta(\mathbf{X}_{t-1})$.

Since

$$
R_{\mathbf{X},x^{(i)}}[f_\theta] = \int d\mathbf{X}_{t-1} P(\mathbf{X}_{t-1}) F(f_\theta(\mathbf{X}_{t-1}))
$$

The minimum of the risk $R_{\mathbf{X},x^{(i)}}[f_\theta]$ is attained iff $F(f_\theta(\mathbf{X}_{t-1}))$ attains minimum at every $\mathbf{X}_{t-1}$, i.e.,

$$
f_\theta(\mathbf{X}_{t-1}) = \int dx_t^{(i)} P(x_t^{(i)}|\mathbf{X}_{t-1}) x_t^{(i)}
$$

is true for any $\mathbf{X}_{t-1}$. Given Assumption 1, we know that $f_\theta \in \mathcal{F}$ has enough capacity such that it can approximate any $\int dx_t^{(i)} P(x_t^{(i)}|\mathbf{X}_{t-1}) x_t^{(i)}$. Therefore,

$$
\operatorname{argmin}_{f_\theta} R_{\mathbf{X},x^{(i)}}[f_\theta] = \int dx_t^{(i)} P(x_t^{(i)}|\mathbf{X}_{t-1}) x_t^{(i)}
$$

and

$$
\min_{f_\theta} R_{\mathbf{X},x^{(i)}}[f_\theta] = \mathbb{E}_{\mathbf{X}_{t-1},x_t^{(i)}} \left[ \left( x_t^{(i)} - \int dx_t^{(i)} P(x_t^{(i)}|\mathbf{X}_{t-1}) x_t^{(i)} \right)^2 \right]
$$

∎

## B.2 MINIMUM MSE WITH DIFFERENT VARIABLES

**Lemma 1.2.** *Suppose that Assumption 1 holds, and $X_{t-1}^{(U)}, X_{t-1}^{(V)}, X_{t-1}^{(W)} \subset \mathbf{X}_{t-1}$ are mutually exclusive sets of variables satisfying*

$$X_{t-1}^{(W)} \perp\!\!\!\perp x_t^{(i)} | X_{t-1}^{(U)}, X_{t-1}^{(V)}, \qquad X_{t-1}^{(V)} \not\perp\!\!\!\perp x_t^{(i)} | X_{t-1}^{(U)}, X_{t-1}^{(W)}$$

*Then*

$$min_{f_\theta} \mathbb{E}_{X_{t-1}^{(U)}, X_{t-1}^{(V)}, x_t^{(i)}} \left[ \left( x_t^{(i)} - f_\theta(X_{t-1}^{(U)}, X_{t-1}^{(V)}) \right)^2 \right] < min_{f_\theta} \mathbb{E}_{X_{t-1}^{(U)}, X_{t-1}^{(V)}, x_t^{(i)}} \left[ \left( x_t^{(i)} - f_\theta(X_{t-1}^{(U)}, X_{t-1}^{(W)}) \right)^2 \right]$$

*Proof.* Since Assumption 1 holds, according to Lemma 1.1, Lemma 1.2 is equivalent to

$$\mathbb{E}_{X_{t-1}^{(U)}, X_{t-1}^{(V)}, x_t^{(i)}} \left[ \left( x_t^{(i)} - \int dx_t^{(i)} P(x_t^{(i)} | X_{t-1}^{(U)}, X_{t-1}^{(V)}) x_t^{(i)} \right)^2 \right]$$

$$< \mathbb{E}_{X_{t-1}^{(U)}, X_{t-1}^{(W)}, x_t^{(i)}} \left[ \left( x_t^{(i)} - \int dx_t^{(i)} P(x_t^{(i)} | X_{t-1}^{(U)}, X_{t-1}^{(W)}) x_t^{(i)} \right)^2 \right]$$

We have

$$\mathbb{E}_{X_{t-1}^{(U)}, X_{t-1}^{(W)}, x_t^{(i)}} \left[ \left( x_t^{(i)} - \int dx_t^{(i)} P(x_t^{(i)} | X_{t-1}^{(U)}, X_{t-1}^{(W)}) x_t^{(i)} \right)^2 \right]$$

$$= \int dX_{t-1}^{(U)} dX_{t-1}^{(W)} dx_t^{(i)} P(X_{t-1}^{(U)}, X_{t-1}^{(W)}, x_t^{(i)}) \left( x_t^{(i)} - \int dx_t^{(i)} P(x_t^{(i)} | X_{t-1}^{(U)}, X_{t-1}^{(W)}) x_t^{(i)} \right)^2$$

$$= \int dX_{t-1}^{(U)} dX_{t-1}^{(V)} dX_{t-1}^{(W)} dx_t^{(i)} P(X_{t-1}^{(U)}, X_{t-1}^{(V)}, X_{t-1}^{(W)}, x_t^{(i)}) \left( x_t^{(i)} - \int dx_t^{(i)} P(x_t^{(i)} | X_{t-1}^{(U)}, X_{t-1}^{(W)}) x_t^{(i)} \right)^2$$

$$= \int dX_{t-1}^{(U)} dX_{t-1}^{(V)} dX_{t-1}^{(W)} P(X_{t-1}^{(U)}, X_{t-1}^{(V)}) P(X_{t-1}^{(W)} | X_{t-1}^{(U)}, X_{t-1}^{(V)}) \cdot$$

$$\int dx_t^{(i)} P(x_t^{(i)} | X_{t-1}^{(U)}, X_{t-1}^{(V)}) \left( x_t^{(i)} - \int dx_t^{(i)} P(x_t^{(i)} | X_{t-1}^{(U)}, X_{t-1}^{(W)}) x_t^{(i)} \right)^2$$

$$> \int dX_{t-1}^{(U)} dX_{t-1}^{(V)} dX_{t-1}^{(W)} P(X_{t-1}^{(U)}, X_{t-1}^{(V)}) P(X_{t-1}^{(W)} | X_{t-1}^{(U)}, X_{t-1}^{(V)}) \cdot$$

$$\int dx_t^{(i)} P(x_t^{(i)} | X_{t-1}^{(U)}, X_{t-1}^{(V)}) \left( x_t^{(i)} - \int dx_t^{(i)} P(x_t^{(i)} | X_{t-1}^{(U)}, X_{t-1}^{(V)}) x_t^{(i)} \right)^2$$

$$= \int dX_{t-1}^{(U)} dX_{t-1}^{(V)} P(X_{t-1}^{(U)}, X_{t-1}^{(V)}) \int dx_t^{(i)} P(x_t^{(i)} | X_{t-1}^{(U)}, X_{t-1}^{(V)}) \left( x_t^{(i)} - \int dx_t^{(i)} P(x_t^{(i)} | X_{t-1}^{(U)}, X_{t-1}^{(V)}) x_t^{(i)} \right)^2$$

$$= \mathbb{E}_{X_{t-1}^{(U)}, X_{t-1}^{(V)}, x_t^{(i)}} \left[ \left( x_t^{(i)} - \int dx_t^{(i)} P(x_t^{(i)} | X_{t-1}^{(U)}, X_{t-1}^{(V)}) x_t^{(i)} \right)^2 \right]$$

The third equality (the one before the inequality) is due to that $X_{t-1}^{(W)} \perp\!\!\!\perp x_t^{(i)} | X_{t-1}^{(U)}, X_{t-1}^{(V)}$, leading to $P(X_{t-1}^{(U)}, X_{t-1}^{(V)}, X_{t-1}^{(W)}, x_t^{(i)}) = P(X_{t-1}^{(U)}, X_{t-1}^{(V)}) P(X_{t-1}^{(W)} | X_{t-1}^{(U)}, X_{t-1}^{(V)}) P(x_t^{(i)} | X_{t-1}^{(U)}, X_{t-1}^{(V)})$. The inequality step first uses the setting in Eq. (1) that the noise variables $u_i$ are effective arguments of the response functions $h_i$, and that each $h_i$ is "causality in mean". Therefore, $\int dx_t^{(i)} P(x_t^{(i)} | X_{t-1}^{(U)}, X_{t-1}^{(V)}) x_t^{(i)} \neq \int dx_t^{(i)} P(x_t^{(i)} | X_{t-1}^{(U)}, X_{t-1}^{(W)}) x_t^{(i)}$. Using Lemma 1.1, we have $f_\theta(X_{t-1}^{(U)}, X_{t-1}^{(V)}) = \int dx_t^{(i)} P(x_t^{(i)} | X_{t-1}^{(U)}, X_{t-1}^{(V)}) x_t^{(i)}$ minimizes $\int dx_t^{(i)} P(x_t^{(i)} | X_{t-1}^{(U)}, X_{t-1}^{(V)}) \left( x_t^{(i)} - f_\theta(X_{t-1}^{(U)}, X_{t-1}^{(V)}) \right)^2$, hence the inequality. ∎

**Lemma 1.3.** *Suppose that Assumption 1 holds, and $X_{t-1}^{(D)} \subseteq \mathbf{X}_{t-1}$ are the set of variables that directly structurally causes $x_t^{(i)}$. Then $\forall X_{t-1}^{(S)} \subseteq \mathbf{X}_{t-1}$ with $X_{t-1}^{(S)} \neq X_{t-1}^{(D)}$, we have*

$$min_{f_\theta} \mathbb{E}_{X_{t-1}^{(D)}} \left[ \left( x_t^{(i)} - f_\theta(X_{t-1}^{(D)}) \right)^2 \right] < min_{f_\theta} \mathbb{E}_{X_{t-1}^{(S)}} \left[ \left( x_t^{(i)} - f_\theta(X_{t-1}^{(S)}) \right)^2 \right]$$

*Specifically, we have*

$$min_{f_\theta} \mathbb{E}_{X_{t-1}^{(D)}} \left[ \left( x_t^{(i)} - f_\theta(X_{t-1}^{(D)}) \right)^2 \right] < min_{f_\theta} \mathbb{E}_{X_{t-1}^{(\hat{D})}} \left[ \left( x_t^{(i)} - f_\theta(X_{t-1}^{(\hat{D})}) \right)^2 \right]$$

*where $X_{t-1}^{(\hat{D})} = \mathbf{X}_{t-1} \backslash X_{t-1}^{(D)}$.*

*Proof.* For any $X_{t-1}^{(S)}$, let $X_{t-1}^{(U)} = X_{t-1}^{(D)} \cap X_{t-1}^{(S)}$, $X_{t-1}^{(V)} = X_{t-1}^{(D)} \backslash X_{t-1}^{(S)}$, $X_{t-1}^{(W)} = X_{t-1}^{(S)} \backslash X_{t-1}^{(D)}$. Then $X_{t-1}^{(U)}, X_{t-1}^{(V)}, X_{t-1}^{(W)}$ are mutually exclusive, and $X_{t-1}^{(D)} = X_{t-1}^{(U)} \cup X_{t-1}^{(V)}$, $X_{t-1}^{(S)} = X_{t-1}^{(U)} \cup X_{t-1}^{(W)}$. Now we prove that $\forall X_{t-1}^{(S)} \subseteq \mathbf{X}_{t-1}$ with $X_{t-1}^{(S)} \neq X_{t-1}^{(D)}$, the corresponding $X_{t-1}^{(U)}, X_{t-1}^{(V)}, X_{t-1}^{(W)}, x_t^{(i)}$ satisfy the condition for Lemma 1.2. Since $X_{t-1}^{(D)}$ are the set of variables that directly structurally causes $x_t^{(i)}$, there does not exist a $X_{t-1}^{(S)}$ such that the corresponding $X_{t-1}^{(V)} \perp\!\!\!\perp x_t^{(i)} | X_{t-1}^{(U)}, X_{t-1}^{(W)}$ (otherwise it violates the definition of direct structural causality). Thus $X_{t-1}^{(V)} \not\perp\!\!\!\perp x_t^{(i)} | X_{t-1}^{(U)}, X_{t-1}^{(W)}$. To prove $X_{t-1}^{(W)} \perp\!\!\!\perp x_t^{(i)} | X_{t-1}^{(U)}, X_{t-1}^{(V)}$, note that $X_{t-1}^{(W)}$ does not directly structurally cause $x_t^{(i)}$, then by Theorem 1, $X_{t-1}^{(W)}$ does not Granger-cause $x_t^{(i)}$, i.e. $P(x_t^{(i)} | X_{t-1}^{(U)}, X_{t-1}^{(V)}) = P(x_t^{(i)} | X_{t-1}^{(U)}, X_{t-1}^{(V)}, X_{t-1}^{(W)})$, which is equivalent to $X_{t-1}^{(W)} \perp\!\!\!\perp x_t^{(i)} | X_{t-1}^{(U)}, X_{t-1}^{(V)}$. The special case of $X_{t-1}^{(\hat{D})}$ follows directly that $X_{t-1}^{(\hat{D})} = \mathbf{X}_{t-1} \backslash X_{t-1}^{(D)} \neq X_{t-1}^{(D)}$ and letting $X_{t-1}^{(S)} = X_{t-1}^{(\hat{D})}$. ∎

### B.3 QUALITATIVE AND QUANTITATIVE BEHAVIORS OF THE LEARNABLE NOISE RISK

In this section, we analyze the qualitative and quantitative behaviors of the learnable noise risk (Eq. 2), with varying noise levels $\eta_j$. For each variable $X_{t-1}^{(j)} \in \mathbf{X}_{t-1}$, $j = 1, 2, ...N$, define $\rho_j = \tanh\left( I(X_{t-1}^{(j)}; \tilde{X}_{t-1}^{(j)(\eta_j)}) \right) \in [0, 1]$ as a "rescaled" mutual information between $X_{t-1}^{(j)}$ and $\tilde{X}_{t-1}^{(j)(\eta_j)}$. When $\eta_j = \mathbf{0}$ so that $\tilde{X}_{t-1}^{(j)(\eta_j)} = X_{t-1}^{(j)}$, $\rho_j = 1$. When all elements of $\eta_j \to \infty$, $\rho_j = 0$. Denoting $\boldsymbol{\rho} = (\rho_1, \rho_2, ...\rho_N)$, we can then rewrite the learnable noise risk (Eq. 2) as

$$R_{\mathbf{X}, x^{(i)}}[f_\theta, \boldsymbol{\rho}] = \text{MMSE}^{(i)}(\boldsymbol{\rho}) + \lambda \cdot \sum_{j=1}^{N} \text{arctanh}(\rho_j) \tag{9}$$

where $\text{MMSE}^{(i)}(\boldsymbol{\rho}) = \min_{\boldsymbol{\eta}, f_\theta} \mathbb{E}_{\mathbf{X}_{t-1}, x_t^{(i)}, \boldsymbol{\epsilon}} \left[ \left( x_t^{(i)} - f_\theta(\tilde{\mathbf{X}}_{t-1}^{(\boldsymbol{\eta})}) \right)^2 \right]$ subject to $\rho_j = \tanh\left( I(X_{t-1}^{(j)}; \tilde{X}_{t-1}^{(j)(\eta_j)}) \right), j = 1, 2, ...N$. Let $X_{t-1}^{(D)} \subseteq \mathbf{X}_{t-1}$ be the set of variables that directly structurally causes $x_t^{(i)}$, and denote the corresponding set of $\rho_j$ as $\boldsymbol{\rho}^{(D)}$. Denote $X_{t-1}^{(\hat{D})} = \mathbf{X}_{t-1} \backslash X_{t-1}^{(D)}$ and the corresponding set of $\rho_j$ as $\boldsymbol{\rho}^{(\hat{D})}$. For any $i = 1, 2, ...N$, we have the following properties:

1. $\text{MMSE}^{(i)}(\boldsymbol{\rho})$ attains maximum at $\boldsymbol{\rho} = \mathbf{0}$.

2. $\text{MMSE}^{(i)}(\boldsymbol{\rho})$ is monotonically decreasing w.r.t. each $\rho_j$.

3. $\text{MMSE}^{(i)}(\boldsymbol{\rho})\big|_{\boldsymbol{\rho}^{(D)}=\mathbf{1}, \boldsymbol{\rho}^{(\hat{D})}=\mathbf{0}} < \text{MMSE}^{(i)}(\boldsymbol{\rho})\big|_{\boldsymbol{\rho}^{(D)}=\mathbf{0}, \boldsymbol{\rho}^{(\hat{D})}=\mathbf{1}}$ (using Lemma 1.3).

4. $\text{MMSE}^{(i)}(\boldsymbol{\rho})$ attains minimum at $\boldsymbol{\rho}^{(D)} = \mathbf{1}$. $\text{MMSE}^{(i)}(\boldsymbol{\rho})\big|_{\boldsymbol{\rho}^{(D)}=\mathbf{1}}$ is constant w.r.t. $\boldsymbol{\rho}^{(\hat{D})}$.

To get a better intuition of the landscape of $R_{\mathbf{X},x^{(i)}}[f_\theta, \boldsymbol{\rho}]$, let's investigate a simple example. Let the response function be:

$$\begin{cases} x_t^{(1)} := h_1(u_1) = \sqrt{\Sigma_x} \cdot u_1 \\ x_t^{(2)} := h_2(x_{t-1}^{(1)}, u_2) = x_{t-1}^{(1)} + \sqrt{\Omega_x} \cdot u_2 \\ x_t^{(3)} := h_3(x_{t-1}^{(2)}, u_3) = x_{t-1}^{(2)} + \sqrt{\Omega_y} \cdot u_3 \end{cases} \tag{10}$$

where $u_1, u_2, u_3$ are independent unit Gaussian variables, and $\mathbf{X}_{t-1} = (X_{t-1}^{(1)}, X_{t-1}^{(2)}, X_{t-1}^{(3)}) = \left( (x_{t-2}^{(1)}, x_{t-1}^{(1)}), (x_{t-2}^{(2)}, x_{t-1}^{(2)}), (x_{t-2}^{(3)}, x_{t-1}^{(3)}) \right)$. For $R_{\mathbf{X},x^{(3)}}[f_\theta, \boldsymbol{\rho}] = \text{MMSE}^{(3)}(\boldsymbol{\rho}) + \lambda \cdot \sum_{j=1}^{3} \text{arctanh}(\rho_j)$, since only $x_{t-2}^{(1)}$ and $x_{t-1}^{(2)}$ are d-connected to $x_t^{(3)}$, at the minimization of $R_{\mathbf{X},x^{(3)}}[f_\theta, \boldsymbol{\rho}]$, only $x_{t-2}^{(1)}$ and $x_{t-1}^{(2)}$ may have a finite $\eta_{j,l}^*$ (the other $\eta_{j,l}^*$ are all infinite). Therefore, setting the $\eta_{j,l}$ not corresponding to $x_{t-2}^{(1)}$ and $x_{t-1}^{(2)}$ as infinity, and let $\tilde{x}_{t-2}^{(1)} = x_{t-2}^{(1)} + \eta_x \cdot \epsilon_x$, $\tilde{x}_{t-1}^{(2)} = x_{t-1}^{(2)} + \eta_y \cdot \epsilon_y$, $\epsilon_x$ and $\epsilon_x$ being independent unit Gaussian variables. Let $f_\theta(x_{t-2}^{(1)}, x_{t-1}^{(2)}) = a \cdot x_{t-2}^{(1)} + b \cdot x_{t-1}^{(2)}$, then we can get an analytic expression for $R_{\mathbf{X},x^{(3)}}[f_\theta, \eta_x, \eta_y]$:

$$R_{\mathbf{X},x^{(3)}}[f_\theta, \eta_x, \eta_y]$$
$$= a^2 \Sigma_x + (b-1)^2 (\Sigma_x + \Omega_x) + a^2 \eta_x^2 + b^2 \eta_y^2 + 2a(b-1)\Sigma_x + \Omega_y + \frac{\lambda}{2}\log\left(1 + \frac{\Sigma_x}{\eta_x^2}\right) + \frac{\lambda}{2}\log\left(1 + \frac{\Sigma_x + \Omega_x}{\eta_y^2}\right)$$

Minimizing $R_{\mathbf{X},x^{(3)}}[f_\theta, \eta_x, \eta_y]$ w.r.t. $a$ and $b$, we get

$$a^* = \frac{\eta_y^2 \Sigma_x}{\eta_x^2 \eta_y^2 + \eta_x^2 \Sigma_x + \eta_y^2 \Sigma_x + \eta_x^2 \Omega_x + \Omega_x \Sigma_x}$$

$$b^* = \frac{\eta_x^2(\Sigma_x + \Omega_x) + \Sigma_x \Omega_x}{\eta_x^2 \eta_y^2 + \eta_x^2 \Sigma_x + \eta_y^2 \Sigma_x + \eta_x^2 \Omega_x + \Omega_x \Sigma_x}$$

Substituting into $R_{\mathbf{X},x^{(3)}}[f_\theta, \eta_x, \eta_y]$, we have

$$R_{\mathbf{X},x^{(3)}}[\eta_x, \eta_y]$$
$$= \min_{f_\theta} R_{\mathbf{X},x^{(3)}}[f_\theta, \eta_x, \eta_y]$$
$$= \frac{\eta_y^2(\Sigma_x \Omega_x + \eta_x^2(\Sigma_x + \Omega_x))}{\eta_x^2 \eta_y^2 + \eta_x^2 \Sigma_x + \eta_y^2 \Sigma_x + \eta_x^2 \Omega_x + \Omega_x \Sigma_x} + \frac{\lambda}{2}\log\left(1 + \frac{\Sigma_x}{\eta_x^2}\right) + \frac{\lambda}{2}\log\left(1 + \frac{\Sigma_x + \Omega_x}{\eta_y^2}\right)$$

Here we have neglected the constant $\Omega_y$. To obtain $R_{\mathbf{X},x^{(3)}}[\boldsymbol{\rho}]$, let $\rho_1 = \tanh\left(\frac{1}{2}\log\left(1 + \frac{\Sigma_x}{\eta_x^2}\right)\right)$, $\rho_2 = \tanh\left(\frac{1}{2}\log\left(1 + \frac{\Sigma_x + \Omega_x}{\eta_x^2}\right)\right)$, we have $\eta_x^2 = \frac{1-\rho_1}{2\rho_1}\Sigma_x$, $\eta_y^2 = \frac{1-\rho_2}{2\rho_2}(\Sigma_x + \Omega_x)$. Substituting, we have

$$R_{\mathbf{X},x^{(3)}}[\boldsymbol{\rho}] = \text{MMSE}^{(3)}(\boldsymbol{\rho}) + \lambda \cdot \sum_{j=1}^{2} \text{arctanh}(\rho_j)$$

$$= \frac{(\rho_2 - 1)(\Sigma_x + \Omega_x)((\rho_1 - 1)\Sigma_x - (\rho_1 + 1)\Omega_x)}{(1 + \rho_1 + \rho_2 - 3\rho_1\rho_2)\Sigma_x + (1 + \rho_1)(1 + \rho_2)\Omega_x} + \lambda \cdot \text{arctanh}(\rho_1) + \lambda \cdot \text{arctanh}(\rho_2)$$

Fig. 5 shows the landscape of $\text{MMSE}^{(3)}(\boldsymbol{\rho})$ and $R_{\mathbf{X},x^{(3)}}[\boldsymbol{\rho}]$, for $\Sigma_x = 1, \Omega_x = 2, \lambda = 1$. We see that $\text{MMSE}^{(3)}(\boldsymbol{\rho})$ satisfies the above mentioned four properties. Particularly, $\text{MMSE}^{(3)}(\boldsymbol{\rho})\big|_{\rho_1=1,\rho_2=0} > \text{MMSE}^{(3)}(\boldsymbol{\rho})\big|_{\rho_1=0,\rho_2=1}$. After adding $\lambda \cdot \text{arctanh}(\rho_1) + \lambda \cdot \text{arctanh}(\rho_2)$, the $R_{\mathbf{X},x^{(3)}}[\boldsymbol{\rho}]$ has global minimum along $\rho_1 = 0$ largely due to this property. Therefore, for this particular example, when $R_{\mathbf{X},x^{(3)}}[\boldsymbol{\rho}]$ is minimized, $\rho_1 = 0$, i.e. $I(x_{t-2}^{(1)}, \tilde{x}_{t-2}^{(1)(\eta_1^*)}) = 0$.

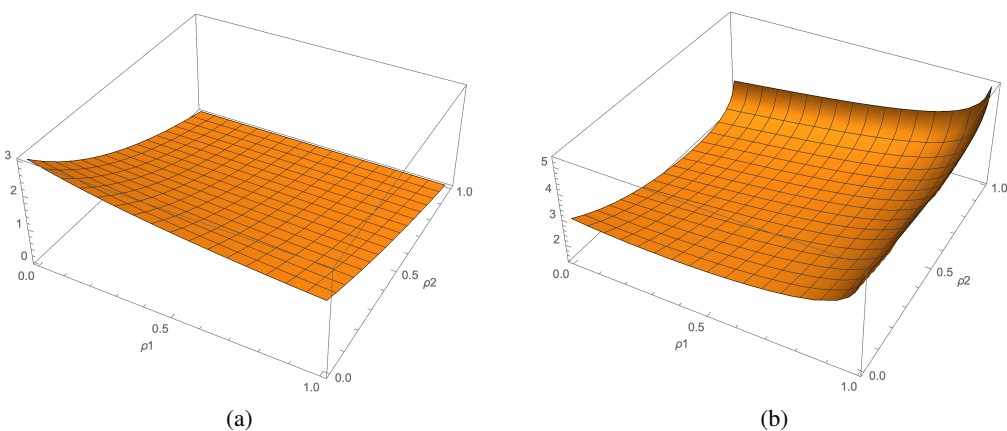

(a)          (b)

Figure 5: (a) MMSE$^{(3)}(\boldsymbol{\rho})$ and (b) $R_{\mathbf{X},x^{(3)}}[\boldsymbol{\rho}]$ in section B.3, for $\Sigma_x = 1, \Omega_x = 2, \lambda = 1$.

By varying the value of $\lambda$, we can tune the relative influence of the two terms MMSE$^{(3)}(\boldsymbol{\rho})$ and $\sum_{j=1}^{2} \operatorname{arctanh}(\rho_j)$. The landscape corresponding to $\lambda = 0.01, 0.5, 2, 10$ are plotted in Fig. 6. We see that when $\lambda \ll 1$, the MMSE term dominates, and it is possible that the global minimum of $R_{\mathbf{X},x^{(3)}}[\boldsymbol{\rho}]$ is not at $\rho_1 = 0$. This is similar to the effect of a L1 regularization, where if the coefficient $\lambda$ for the L1 is vanishingly small, the L1 regularization will barely influence the loss landscape. When $\lambda$ is not vanishingly small, as in Fig. 6 (b), we see that the global minimum of $R_{\mathbf{X},x^{(3)}}[\boldsymbol{\rho}]$ lies on $\rho_1 = 0$. When $\lambda \to +\infty$, the $\sum_{j=1}^{2} \operatorname{arctanh}(\rho_j)$ term dominates and the global minimum is at $\rho_1 = 0, \rho_2 = 0$.

In general, we expect $R_{\mathbf{X},x^{(i)}}[\boldsymbol{\rho}]$ behave qualitatively similar. When $\lambda \to +\infty$, the global minimum for $R_{\mathbf{X},x^{(i)}}[\boldsymbol{\rho}]$ is at $\boldsymbol{\rho}^* = \mathbf{0}$. As we ramp down $\lambda$, the dimension that has largest influence on MMSE will first host the global minimum with nonzero $\rho_j^*$, which is most likely the variable that directly structurally causes $x_i^{(i)}$. When $\lambda$ is further ramping down, we expect that the variables that host the global minimum with nonzero $\rho_j$ will more likely be those that directly structurally causes $x_i^{(i)}$, due to the landscape influenced by the four properties of MMSE. This can justify the learnable noise risk as a good objective for causal discovery/variable selection. The experiments in the paper will empirically test the performance of the learnable noise risk.

## C    UPPER BOUND FOR THE LEARNABLE NOISE RISK

In this section, we prove that $I(\tilde{X}_{t-1}^{(j)(\eta_j)}; X_{t-1}^{(j)}) \leq \frac{1}{2} \sum_{l=1}^{KM} \log\left(1 + \frac{\operatorname{Var}(X_{t-1,l}^{(j)})}{\eta_{j,l}^2}\right)$. We formally state the theorem as follows:

**Theorem 2.** *Let* $\tilde{X}_{t-1}^{(j)(\eta_j)} := X_{t-1}^{(j)} + \eta_j \cdot \epsilon_j$, $j = 1, 2, ...N$ *be the noise-corrupted inputs with learnable noise amplitudes* $\eta_j \in R^{KM}$, *and* $\epsilon_j \sim N(\mathbf{0}, \mathbf{I})$. *We have*

$$I(\tilde{X}_{t-1}^{(j)(\eta_j)}; X_{t-1}^{(j)}) \leq \frac{1}{2} \sum_{l=1}^{KM} \log\left(1 + \frac{Var(X_{t-1,l}^{(j)})}{\eta_{j,l}^2}\right) \qquad (11)$$

*where l is the $l^{th}$ element of a vector,* $std(X_{t-1,l}^{(j)})$ *is the standard deviation of* $X_{t-1,l}^{(j)}$ *across t. The equality is reached when* $X_{t-1}^{(j)}$ *obeys a multivariate Gaussian distribution with diagonal covariance matrix* $\Sigma$ *satisfying* $\Sigma_{l,l} = Var(X_{t-1,l}^{(j)}) + \eta_{j,l}^2$.

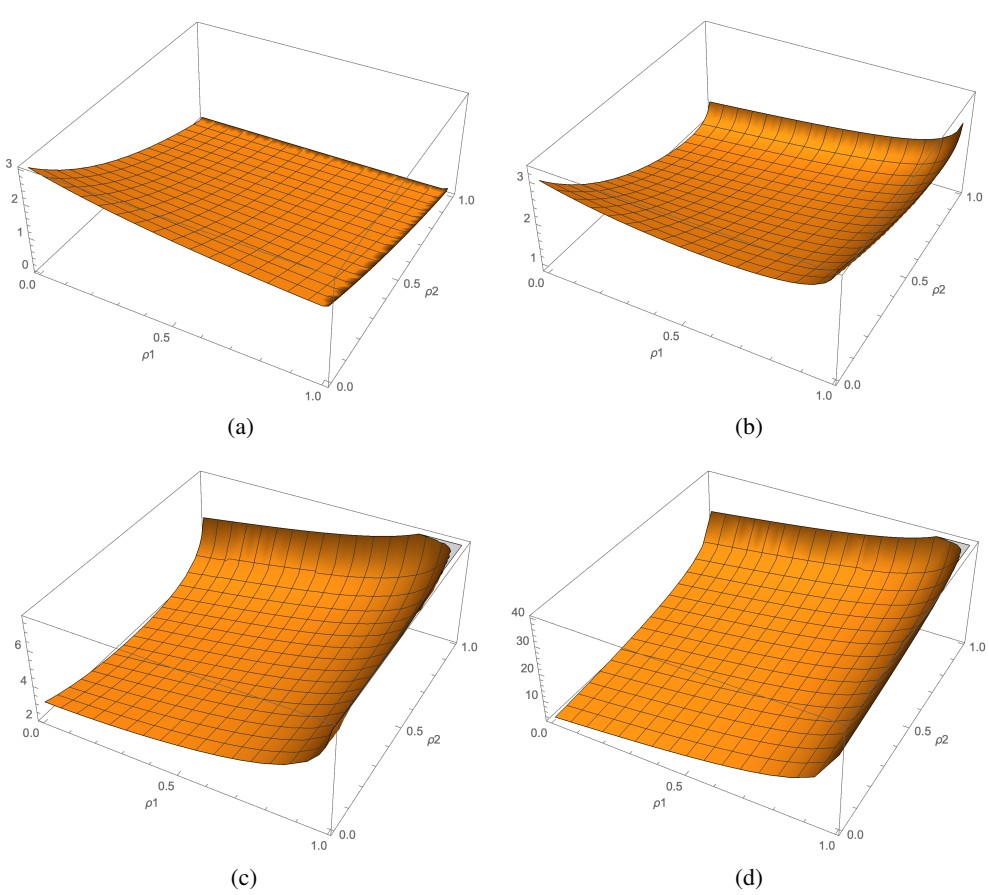

Figure 6: (a) $R_{\mathbf{X},x^{(3)}}[\boldsymbol{\rho}]$ for (a) $\lambda = 0.01$, (b) $\lambda = 0.5$, (c) $\lambda = 2$ and (d) $\lambda = 10$ in section B.3, for $\Sigma_x = 1, \Omega_x = 2$.

*Proof.* We have

$$I(\tilde{X}_{t-1}^{(j)(\eta_j)}; X_{t-1}^{(j)}) = H(\tilde{X}_{t-1}^{(j)(\eta_j)}) - H(\eta_j \cdot \epsilon_j)$$

$$= H(\tilde{X}_{t-1}^{(j)(\eta_j)}) - \left( \frac{KM}{2}\log(2\pi e) + \sum_{l=1}^{KM} \frac{1}{2}\log(\eta_{j,l}^2) \right)$$

Here $H(\cdot)$ is differential entropy. For $\tilde{X}_{t-1}^{(j)(\eta_j)}$, its variance at the $l^{\text{th}}$ dimension is

$$\text{Var}(\tilde{X}_{t-1,l}^{(j)(\eta_j)}) = \text{Var}(X_{t-1,l}^{(j)} + \eta_j \cdot \epsilon_j)$$

$$= \text{Var}(X_{t-1,l}^{(j)}) + \text{Var}(\eta_{j,l} \cdot \epsilon_{j,l})$$

$$= \text{Var}(X_{t-1,l}^{(j)}) + \eta_{j,l}^2$$

The second equality is due to that $X_{t-1}^{(j)}$ is independent of $\epsilon_j$. Using the principle of maximum entropy, the distribution that maximizes $H(\tilde{X}_{t-1}^{(j)(\eta_j)})$ subject to the constraint of $\text{Var}(\tilde{X}_{t-1,l}^{(j)(\eta_j)}) = \text{Var}(X_{t-1,l}^{(j)}) + \eta_{j,l}^2, l = 1, 2, ...KM$ is a Gaussian distribution whose diagonal covariance matrix $\Sigma$ satisfies $\Sigma_{l,l} = \text{Var}(X_{t-1,l}^{(j)}) + \eta_{j,l}^2$. Its entropy is $H(\tilde{X}_{t-1}^{(j)(\eta_j)}) = \frac{KM}{2}\log(2\pi e) + \sum_{l=1}^{KM} \frac{1}{2}\log(\eta_{j,l}^2 + \text{Var}(X_{t-1,l}^{(j)}))$. Therefore,

$$I(\tilde{X}_{t-1}^{(j)(\eta_j)}; X_{t-1}^{(j)})$$

$$\leq \left( \frac{KM}{2}\log(2\pi e) + \sum_{l=1}^{KM} \frac{1}{2}\log(\eta_{j,l}^2 + \text{Var}(X_{t-1,l}^{(j)})) \right) - \left( \frac{KM}{2}\log(2\pi e) + \sum_{l=1}^{KM} \frac{1}{2}\log(\eta_{j,l}^2) \right)$$

$$= \frac{1}{2} \sum_{l=1}^{KM} \log \left( 1 + \frac{\text{Var}(X_{t-1,l}^{(j)})}{\eta_{j,l}^2} \right)$$

The equality is reached when $X_{t-1}^{(j)}$ obeys a multivariate Gaussian distribution with diagonal covariance matrix $\Sigma$ satisfying $\Sigma_{l,l} = \text{Var}(X_{t-1,l}^{(j)}) + \eta_{j,l}^2$. ∎

## D    IMPLEMENTATION DETAILS FOR THE METHODS

Here we state the implementation details for our method, as well as other methods being compared. Throughout this paper, unless otherwise specified, we use the standard technique in Kraskov et al. (2004) to estimate the KL-divergence and mutual information, which is used in our implementations of Mutual information, Transfer Entropy and Causal Influence.

### D.1    OUR METHOD

Without stating otherwise, our method (Algorithm 1) as a default uses a three layer neural net, with two hidden layers having 8 neurons and SELU (Klambauer et al. (2017)) activation, and the last layer having linear activation. Adam (Kingma & Ba (2014)) optimizer with learning rate $= 10^{-4}$ is used as default throughout this paper. We set $\eta_0 = 0.01$ and $\lambda = 0.01$. We use 10000 epochs, with early-stopping such that if the best validation loss does not go down for 40 monitoring points (we monitor the validation loss every 20 epochs), do early-stop. It also has a 400 epoch warm-up period where the mutual information term is turned off, to allow $f_\theta$ to find a good initial model as a start. We use the approximation $I(\tilde{X}_{t-1}^{(j)(\eta_j^*)}; X_{t-1}^{(j)}) \simeq \frac{1}{2} \sum_{l=1}^{KM} \log(1 + 1/\chi_{j,l}^2)$ in the risk and also in estimating $W_{ji}$, as discussed in the main text in 3.3. Here $\chi_{j,l} = \frac{\eta_{j,l}}{\text{std}(X_{t-1,l}^{(j)})}$ is the relative noise scale w.r.t. the standard deviation of each element, $l$ denoting the $l^{\text{th}}$ element of the $KM$-dimensional vector. In this work, we fix $\chi_{j,l}$ to be the same for each $j$, and let $\chi_j$ be a single parameter instead of a vector. This simplifies the risk calculation, and also to a first order invariant to the reparameterizaiton of each time series $X_{t-1}^{(j)}$.

## D.2 Transfer Entropy

We use the definition of transfer entropy as defined in Schreiber (2000). In that work the transfer entropy is defined for two time series. To deal with multiple time series, we let $X_{t-1}^{(\hat{j})}$ also include other time series, similar to the extension of transfer entropy as in Lizier et al. (2008).

## D.3 Causal Influence

For causal influence, we use the same network architecture as in our method, to learn a prediction model. Then the KL divergence is estimated via the technique in Kraskov et al. (2004).

## D.4 Linear Granger

We use the definition of linear Granger causality (Eq. (7) and (8) in Ding et al. (2006)) to calculate linear Granger causality. Specifically, we estimate the variance of the residue of a linear predictor of $x_{t-1}^{(i)}$ with and without $X_{t-1}^{(j)}$ (also conditioned on $\mathbf{X}_{t-1}^{(\hat{j})}$), using Levinson-Whittle(-Wiggins) and Robinson algorithm (Morf et al. (1978)). Then the linear Granger causality equals the log of the ratio of the two variances.

## E Implementation details for synthetic experiments

For all experiments in this section, each metrics is obtained by performing the experiments (including generation of the dataset and the training) four times with seed = $0, 30, 60, 90$ and averaging the resulting metrics. For the ground-truth causal tensor $A$, each element $A_{ji}$ is a $K \times M$ matrix, with 0.5 probability of being an all-zero matrix, and 0.5 probability of being a nonzero matrix. If $A_{ji}$ is a nonzero matrix, its each element is sampled from a log-normal distribution with $\mu = 0$ and $\sigma = 1$. For $B$, each $B_j$ is also a $K \times N$ matrix, with each element sampling from $U[-1, 1]$. We use $H_1(x) = \mathrm{softplus}(x) = \log(1 + e^x)$, and $H_2(x) = \tanh(x)$ in equation (6). As a default, 500 time series each with length of 22 are generated from Eq. (6), each of which is wrapped into 19 $(\mathbf{X}_{t-1}, x_t^{(i)})$ pairs (since $K = 3$), so there are in total $500 \times 19 = 9500$ examples for each dataset. The train-test-split is 4:1 for all experiments in this paper.

## F Details for the video game dataset

Here, we implement a custom Atari Breakout game in the OpenAI Gym (Brockman et al. (2016)) environment, mimicking the original game, where we can access the state of the ball, paddle and bricks, etc. This representation is also used in the OO-MDP (Diuk et al. (2008)) paradigm for a more efficient representation of the environment state. We use the DQN algorithm, the same CNN architecture as in Mnih et al. (2015) to train an RL agent. Then we let it play the game for 20000 steps, obtaining a dataset with time-length of 20000 steps (if the agent dies, we restart the game) and 6 time series: action, paddle's $x$ position, ball's $x$ position, ball's $y$ position, number of bricks and reward. We then feed the time series (each time series normalized to mean of 0 and variance of 1) to our method, the same procedure as performed in the synthetic experiment, to let it produce an inferred causal matrix $W_{ji}$, which is shown in Fig. 1. All the datasets used in this paper and code will be open-sourced upon publication of the paper.

## G Implementation details for the real-world dataset

For the two real-world datasets, we obtain the data with the same procedure as in Ancona et al. (2004). Then the data (each time series normalized to mean of 0 and variance of 1) are fed into our algorithm to infer the causal strength $W_{ji}$ with the default settings as described in Appendix D and E. For each $K = 1, 2, ...20$, the experiments are run for 50 times with seed from 0 to 49, and Fig. 3 and Fig. 4 are obtained by averaging over the inferred $W$ matrix.

## H Causality results for the rat-EEG dataset by previous works

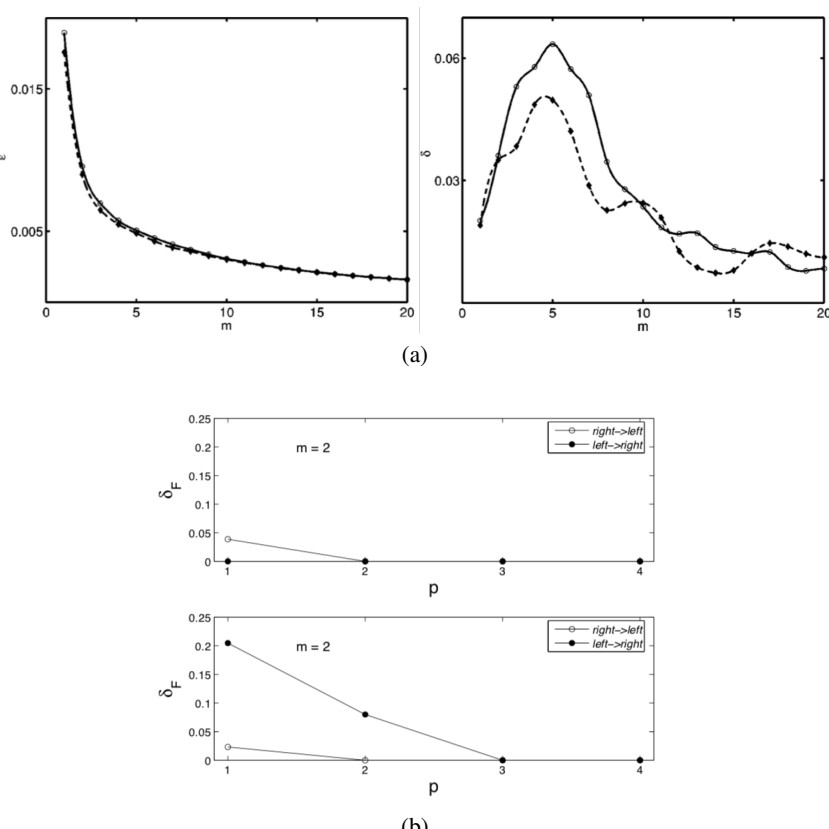

(a)

(b)

Figure 7: Causal indices for the rat EEG dataset with previous methods. (a) By Ancona et al. (2004). Left: the variance for the left EEG (open circles) and right EEG (diamonds) vs. time lag $m$ before brain lesion. Right: the causality index after brain lesion. (b) By Marinazzo et al. (2008b). The filtered causality index vs. varying $p$, the order of the inhomogeneous polynomial kernel, before (upper) and after (lower) brain lesion.

