# OpenReview forum: "Neural Causal Discovery with Learnable Input Noise"
_ICLR.cc/2019/Conference_

### Official Review · AnonReviewer2 · 2018-10-23
**Interesting approach**

**Rating:** 8
**Confidence:** 4

**Review:**

In the manuscript entitled "Neural Causal Discovery with Learnable Input Noise" the authors describe a method for automated causal inference under the scenario of a stream of temporally structured random variables (with no missingness and a look-back window of given size).  The proposed approach combines a novel measure of the importance of fidelty in each variable to predictive accuracy of the future system state ("learnable noise risk") with a flexible functional approximation (neural network).  Although the setting (informative temporal data) is relatively restricted with respect to the general problem of causal inference, this is not unreasonable given the proposed direction of application to automated reasoning in machine learning.  The simulation and real data experiments are interesting and seem well applied.

A concern I have is that the manuscript as it stands is positioned somewhere between two distinct fields (sparse learning/feature selection, and causal inference for counterfactual estimation/decision making), but doesn't entirely illustrate its relationship to either.  In particular, the derived criterion is comparable to other sparsity-inducing penalities on variable inclusion in machine learning models; although it has motivation in causality it is not exclusively derived from this position, so one might wonder how alternative sparsity penalities might perform on the same challenge.  Likewise, it is not well explained what is the value of the learnt relationships, and how uncertainty and errors in the causal learning are relevant to the downstream use of the learnt model.  In the ordinary feature selection regime one is concerned simply with improving the predictive capacity of models: e.g. a non-linear model might be fit using just the causal variables that might out-perform both a linear model and a non-linear model fit using all variables.  Here the end goal is less clear; this is understandable in the sense that the work is positioned as a piece in a grand objective, but it would seem valuable to nevertheless describe some concrete example(s) to elucidate this aspect of the algorithm (use case / error effects downstream).

---

> ### Author Response · Authors · 2018-11-27
> **Response**
>
> Thank you for the review, and we really appreciate your suggestions!
>
> In the revision, we have added analysis in section 4.2 and section 5 on how the learned causal matrix can be used downstream, for example in RL/IL and interpretability of neural nets. In the discussion in section 5, we also analyze how the error may affect the tasks downstream. We are excited that various tasks may utilize or incorporate our algorithm, and benefit from the causal inference ability it enables.
>
> We have also added comparison with sparse learning/feature selection methods in the “related works” section. In particular, we note that L1 and group L1 regularization is dependent on the model structure change and rescaling of input variables, while our learnable noise risk is invariant to both, making it suitable for causal discovery where the scale of data may span orders of magnitude and the model structure may vary.

---

### Official Review · AnonReviewer3 · 2018-11-05
**A easy-to-follow paper on nonlinear Granger causality which requires some further clarification**

**Rating:** 4
**Confidence:** 4

**Review:**

This paper aims to estimate time-delayed, nonlinear causal influences from time series under the causal sufficiency assumption. It is easy to follow and contains a lot of empirical results. Thanks for the results, but I have several questions.

First, In Theorem 2, which seems to be a main result of the paper, the authors were concerned with the condition when W_{ji} >0, but there is not conclusion if W_{ji} =0. In order to correctly estimate causal relations from data, both cases must be considered.

Second, the conclusion of Theorem 2 seems to be flawed. Let me try to make it clear with the following example. Suppose x^1_{t-2} directly causes x^2_{t-1} and that x^2_{t-1} directly causes x^3_{t}, without a direct influence from x^1_{t-2}  to x^3_{t}. Then when minimizing (2), we have the following results step by step:
1) The noise standard deviation in x^2_{t-1}, denoted by \eta_2, may be non-zero. This is because we minimize a tradeoff of the prediction error (the first term in (2)) and a function of the reciprocal of the noise standard deviation \eta_2 (the second term in (2)), not only the prediction error.
2) If \eta_2 is non-zero, then x^1_{t-2} will be useful for the purpose of predicting x^3_{t}. (Note that if \eta_2 is zero, then x^1_{t-2} is not useful for predicting x^3_{t).) From the d-separation perspective, this is because x^1_{t-2} and x^3_{t} are not d-separated by x^2_{t-1} + \eta_2 \cdot \epsilon_2, although they are d-separated by x^2_{t-1}. Then the causal Markov condition tells use that x^1_{t-2} and x^3_{t} are not independent conditional on x^2_{t-1} + \eta_2 \cdot \epsilon_2, which means that x^1_{t-2} is useful for predicting x^3_{t}.
3) Given that x^1_{t-2} is useful for predicting x^3_{t}, when (2) is minimized, \eta_1 will not go to infinity, resulting in a non-zero W_{13), which *mistakenly* tells us that X^{1}_{t-1} directly structurally causes x^{(3)}_t.

This illustrates that the conclusion of Theorem 2 may be wrong.  I believe this is because the proof of Theorem 2 is flawed in lines 5-6 on Page 16. It does not seem sensible to drop X^{j}_{t-1} + \eta_X \cdot \epsilon_X and attain a smaller value of the cost function at the same time. Please carefully check it, especially the argument given in lines 10-13.

Third, it is rather surprising that the authors didn't mention anything about the traditional causal discovery methods based on conditional independence relations in the data, known as constraint-based methods, such as the PC algorithm (Spirtes et al., 1993), IC algorithm (Pearl, 2000), and FCI (Spirtes et al., 1993). Such methods are directly applicable to time-delayed causal relations by further considering the constraint that effects temporally follow the causes.

Fourth, please make it clear that the proposed method aims to estimate "causality-in-mean" because of the formulation in terms of regression. For instance, if x^j_{t-1} influences only the variance of x^i_{t}, but not its mean, then the proposed method may not detect such a causal influence, although the constraint-based methods can.

Any response would be highly appreciated.

---

> ### Author Response · Authors · 2018-11-27
> **Response**
>
> Thank you very much for the instructive and detailed review!
>
> For the first and second comments, we appreciate the detailed example you proposed. Specifically, we agree with the 1) and 2) of your analysis. For 3), although x^(1)_{t-2} is useful for predicting x^3_{t}, due to the causal chain and the presence of independent noise in the response function Eq. (1), x^(2)_{t-1} is even more useful for predicting x^(3)_{t}. When Eq. (2) is minimized w.r.t. both f_\theta and all \eta, with appropriate \lambda, it is likely that \eta_1 will go to infinity and \eta_2 will be finite, leading to the correct conclusion that X^(1)_{t-1} does not directly structurally cause x^(3)_t. For example, in the new Appendix B.3, we show analytically and numerically that for a linear Gaussian system, the global minimum of the learnable noise risk lies on I(x^(1)_{t-2}; \tilde{x}^(1)_{t-2})=0, i.e. \eta_1->\infty, for a wide range of \lambda.
>
> To study the general landscape and global minimum of the learnable noise risk, we first carefully inspect Theorem 2, and find that its original statement is not true in general. We have replaced the original Theorem 2 with a detailed analysis of the loss landscape of the learnable noise risk. Specifically, there are four properties that the minimum MSE (MMSE, the first term of the learnable noise risk after minimizing w.r.t. f_\theta) must obey, as demonstrated in the new Appendix B. In particular, we prove that the MMSE based only on the uncorrupted variables that directly structurally cause x^(i)_t is the minimum among all MMSE based on any set of uncorrupted variables. These properties will likely lead to nonzero mutual information for the variables that directly structurally cause x^(i)_t, at the global minimum of the learnable noise risk, as we ramp down \lambda from infinity.
>
> In a sense, the learnable noise risk behaves similarly as an L1 regularized risk. Whereas L1 encourages sparsity of the parameters of the model, the mutual information term in the learnable noise risk encourages sparsity of the influence of the inputs, where the strength of sparsity inducing depends on \lambda. As also pointed out in the “related works” in the revision, the learnable noise risk is invariant to model structure change (keeping the function mapping unchanged) and rescaling of inputs, while L1 or group L1 do not, making learnable noise risk suitable for causal discovery where the scale of data may span orders of magnitude and the model structure may vary.
>
> For the third and fourth comment, thanks for pointing out and we have added the constraint-based methods in the related works section, and stressed that we are dealing with “causality in mean” in section 3.1.

---

### Official Review · AnonReviewer1 · 2018-11-07
**An interesting approach ; some concerns regarding assumptions and experiments**

**Rating:** 4
**Confidence:** 5

**Review:**

The paper proposes an approach to learn nonlinear causal relationship from time series data that is based on empirical risk minimization regularized by mutual information.  The mutual information at the minimizer of the objective function  is used as causal measure.
The paper is well written and the proposed method well motivate and intuitive.

However I am concerned by the assumption that the lagged variables X_{t-1}^{(j)} follow a diagonal gaussian distribution. This appears to be very restrictive, since typically the values of time series j at time t-1 are typically depending say of those that time t-2, t-3 etc.

Another key concern concerns scalability.  The authors mention gene regulatory networks , neuroscience etc as key applications. Yet the experiments considered in the paper are limited to very few time series. For instance the simulation experiments use  N=30,  which is much smaller than the number of time series usually involved say in gene regulatory network data.  The real data experiments use N= 6 or N=2. This is way to small.

The real data experiments (sections 4.2 and 4.3) are not very convincing, not only because of the very small size of N, but also because there is no comparison with the other approaches.  How do these compare? Does the proposed approach offer  insights on these datasets which are not captured by the comparison methods?

---

> ### Author Response · Authors · 2018-11-27
> **Response**
>
> Thank you for the instructive review!
>
> Our algorithm 1 minimizes the empirical learnable noise risk (Eq. 4), which does not assume that X_{t-1}^{(j)} follows a diagonal gaussian distribution. Originally, to justify the I^u=1/2 \sum_l log(1+Var(X^(j)_{t-1,l})/ \eta_{j,l}^2) term used in our experiments for estimating mutual information, we used diagonal Gaussian assumption for X_{t-1}^(j) in the experiment. In fact, a better way to justify this is to note that I^u provides an upper bound for the mutual information subject to the constraint of known variance of marginal distributions of X^(j)_{t-1}, and the upper bound is reached with the diagonal Gaussian distribution, as is proved in Appendix C in the revision. Therefore, the assumption of diagonal Gaussian assumption is dropped for the experiments in the revision. Practitioners can choose to optimize an upper bound of the learnable noise risk for better efficiency (as is also used in the experiments in this paper), or use differentiable estimate of mutual information for better accuracy, as has also been pointed out in the paper.
>
> In the revision, we have also added a more detailed comparison with other methods in sections 4.2 and 4.3, showing the strength of our method. For example, in section 4.2, our method correctly identifies important causal arrows, while the four other comparison methods either have more false positives and false negatives, or completely fail to discover causal arrows. In section 4.3, we compare with the results in previous literature. We note that although all compared methods correctly identify the causal relations, our method have the advantage that the inferred causal strength does not decay with increasing history length (we also analyzed that in the original submission).

---

### Meta-Review · Area_Chair1 · 2018-12-17
**unconvincing experiments; original theorem statement incorrect**

**Confidence:** 4
**Recommendation:** Reject

**Metareview:**

Granger Causality is a beautiful operational definition of causality, that reduces causal modeling to the past-to-future predictive strength. The combination of classical granger causality with deep learning is very well motivated as a research problem. As such the continuation of the effort in this paper is strongly encouraged. However, the review process did uncover possible flaws in some of the main, original results of this paper. The reviewers also expressed concerns that the experiments were unconvincing due to very small data sizes. The paper will benefit from a revision and resubmission to another venue, and is not ready for acceptance at ICLR-2019.